# An analytic theory of creativity in convolutional diffusion models

**Mason Kamb** [1]  **Surya Ganguli** [1]

## Abstract

We obtain an analytic, interpretable and predictive theory of creativity in convolutional diffusion models. Indeed, score-matching diffusion models can generate highly original images that lie far from their training data. However, optimal score-matching theory suggests that these models should only be able to produce memorized training examples. To reconcile this theory-experiment gap, we identify two simple inductive biases, locality and equivariance, that: (1) induce a form of combinatorial creativity by preventing optimal score-matching; (2) result in fully analytic, completely mechanistically interpretable, local score (LS) and equivariant local score (ELS) machines that, (3) after calibrating a single time-dependent hyperparameter can quantitatively predict the outputs of trained convolution only diffusion models (like ResNets and UNets) with high accuracy (median $r^2$ of $0.95, 0.94, 0.94, 0.96$ for our top model on CIFAR10, FashionMNIST, MNIST, and CelebA). Our model reveals a *locally consistent patch mosaic* mechanism of creativity, in which diffusion models create exponentially many novel images by mixing and matching different local training set patches at different scales and image locations. Our theory also partially predicts the outputs of pre-trained self-attention enabled UNets (median $r^2 \sim 0.77$ on CIFAR10), revealing an intriguing role for attention in carving out semantic coherence from local patch mosaics.

## 1. Introduction and related work

A deep puzzle of generative AI lies in understanding how it produces apparently creative output: outputs that possess a meaningful relationship to their training data, but are

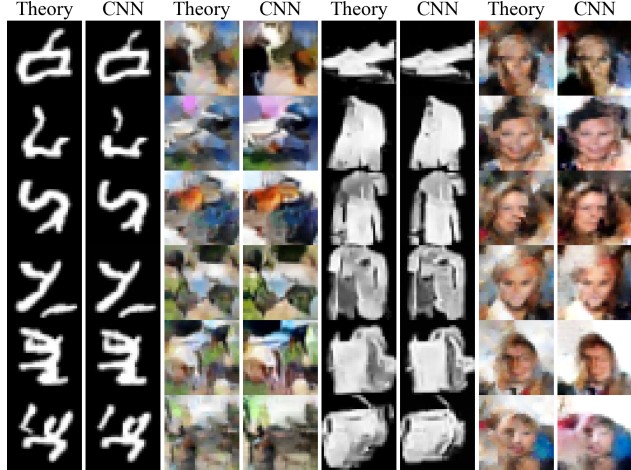

Figure 1. Our analytic theory (left columns) can accurately predict on a *case by case basis* the outputs of convolutional diffusion models (right columns), with UNet or ResNet architectures trained on MNIST, CIFAR10, FashionMNIST, and CelebA (left to right), even when these outputs are highly original and far from the training data. See Fig. 5, App. C, Fig. 10 and Table 2, and App. D, Fig. 13 to Fig. 24 for many more successful theory-experiment comparisons.

nonetheless clearly original, exhibiting novel combinations of attributes that are represented across disparate training examples. What is the nature and origin of this creativity, and how precisely is it generated from a finite training set? We answer these questions for small convolutional diffusion models of images by deriving an analytic and interpretable theory of their behavior that can accurately predict their outputs on a *case-by-case basis* (Fig. 1), and explain how they are created out of *locally consistent patch mosaics* of the training data.

Denoising probabilistic diffusion models (DDPMs) were established in Sohl-Dickstein et al. (2015) and Ho et al. (2020), and then unified with score-matching (Song & Ermon, 2019; Song et al., 2020b). Denoising diffusion implicit models (DDIMs), an alternative deterministic parameterization which we primarily use in this paper, were established in Song et al. (2020a). Diffusion models now play an important role not only in image generation (Dhariwal & Nichol, 2021; Rombach et al., 2022; Ramesh et al., 2022), but also video generation (Ho et al., 2022a;b; Blattmann et al., 2023),

[1]Department of Applied Physics, Stanford University, California, United States. Correspondence to: Mason Kamb <kambm@stanford.edu>, Surya Ganguli <sganguli@stanford.edu>.

*Proceedings of the 42nd International Conference on Machine Learning*, Vancouver, Canada. PMLR 267, 2025. Copyright 2025 by the author(s).

drug design (Alakhdar et al., 2024), protein folding (Watson et al., 2023), and text generation (Li et al., 2023; 2022).

These models are trained to reverse a forward diffusion process that turns the finite training set distribution (a sum of $\delta$-functions over the training points) into an isotropic Gaussian noise distribution, through a time-dependent family of mixtures of Gaussians centered at shrinking data points. Diffusion models are trained to reverse this process by learning and following a score function that points in gradient directions of increasing probability. But therein lies the puzzle of creativity in diffusion models: if the network can learn this *ideal* score function exactly, then they will implement a perfect reversal of the forward process. This, in turn, will *only* be able to turn Gaussian noise into memorized training examples. Thus, any originality in the outputs of diffusion models *must* lie in their *failure* to achieve the very objective they are trained on: learning the ideal score function. But how can they fail in intelligent ways that lead to many sensible new examples *far* from the training set?

Several theoretical and empirical works study the properties of diffusion models. Some works study the sampling properties of these models under the assumption that they learn the ideal score function exactly for a solvable toy class of distributions (Biroli et al., 2024; De Bortoli, 2022; Wang & Vastola, 2023) or up to some small bounded error (Benton et al., 2024). Others establish accuracy guarantees on learning the ideal score function under various assumptions on the data distribution, and the hypothesis class of functions (Lee et al., 2022; Chen et al., 2023; Oko et al., 2023; Ventura et al., 2024; Cui & Zdeborová, 2023; Cui et al., 2023).

As noted above, a key limitation of studying diffusion models under the assumption that they (almost) learn the ideal score function is that such models can only generate memorized training examples; while memorizing behavior has been observed in trained diffusion models (Gu et al., 2023; Somepalli et al., 2023), the ideal score function predicts the model will *always* produce memorized examples, at odds with the creativity of diffusion models in practice. For example, they can compose aspects of their training data in combinatorially many novel ways (Sclocchi et al., 2024; Okawa et al., 2024). This observation has motivated studies of mechanisms behind generalization in diffusion models that underfit the score-matching objective (Kadkhodaie et al., 2023b; Zhang et al., 2023; Wang et al., 2024; Scarvelis et al., 2023). Other works connect creativity in diffusion models to the breakdown of memorization in modern Hopfield networks (Ambrogioni, 2023; Hoover et al., 2023; Pham et al., 2024). However, the extent to which these works can quantitatively predict individual samples from a trained diffusion model on a case-by-case basis is more limited.

To develop theory beyond the memorization regime, we focus on diffusion models with a fully-convolutional backbone, without the self-attention layers introduced in Ho et al. (2020). We identify two fundamental inductive biases that prevent such models from learning the ideal score-function: *translational equivariance*, due to parameter sharing in convolutional layers, and *locality*, due to the model's finite receptive field size. Remarkably, we show these two simple biases are *sufficient* to quantitatively explain the creative outputs of convolutional diffusion models, after calibrating a single time-dependent hyperparameter (the locality scale).

Relatedly, Kadkhodaie et al. (2023a) also identified locality as a limiting constraint in CNN-based diffusion models, but did not attempt to predict their specific outputs. Concurrently with our work, Niedoba et al. (2024) developed a (non-equivariant) patch-based local score approximation model of diffusion models similar to ours, although their quantitative success in predicting the outputs of the neural networks they studied was more limited since they did not study CNNs, which have the strongest locality biases. Another concurrent work (Wang & Vastola, 2024) also studied a Gaussian mixture-based approximation to the reverse process to predict samples on a case-by-case basis. Finally, the results of our analysis exhibit some similarity to very early patch-based texture synthesis methods, e.g. Efros & Leung (1999). Our contributions and outline are as follows:

1. We review why diffusion models that learn the ideal score function can only memorize (Sec. 2).

2. We derive minimum mean squared error (MMSE) approximations to the ideal score function subject to locality, equivariance, and/or partially broken equivariance due to image boundaries. Remarkably, we find simple analytic solutions in all cases (Sec. 3.)

3. These solutions lead to a local score (LS) machine and a boundary-broken equivariant local score (ELS) machine, which constitute fully analytic, mechanistically interpretable theories that can transform noise into creative, structured images without the need for any explicit training process. (Sec. 3).

4. We theoretically characterize samples generated by the ELS machine and show how it achieves *exponential* creativity through *locally consistent patch mosaics* composed of different local training set image patches at different locations in each novel sample (Sec. 4).

5. We show our boundary-broken ELS machine is not only analytic and interpretable but also *predictive*: it can predict, on a case-by-case basis, the outputs of trained UNets and ResNets, achieving median theory-experiment agreements of $r^2 \sim 0.94, 0.95, 0.94, 0.96$ on MNIST, FashionMNIST, CIFAR10, and CelebA for the best architecture on each dataset (Sec. 5). We show that on CelebA32x32, ResNets are best predicted by

the ELS machine (median $r^2 \sim 0.96$), but UNet behavior is better predicted by the fully-local LS machine (median $r^2 \sim 0.90$).

6. Our comparison between theory and experiment reveals that trained diffusion models exhibit a coarse-to-fine generation of spatial structure over time and use image boundaries to anchor image generation (Sec. 5).

7. Our theory reproduces the notorious behavior of diffusion models to generate spatially inconsistent images at fine spatial scales (e.g. incorrect numbers of limbs) and explains its origin in terms of excessive locality at late times in the reverse generative process. (Sec. 5).

8. We compare our purely local ELS machine theory to more powerful trained UNet architectures with non-local self-attention (SA) layers. Our local theory can still partially predict their non-local outputs (median $r^2$ of 0.77 on CIFAR10), but reveal an interesting role for attention in carving out semantically coherent objects from the ELS machine's local patch mosaics (Sec. 6).

Overall our work illuminates the mechanism of creativity in convolutional diffusion models and forms a foundation for studying more powerful attention-enabled counterparts.

## 2. The ideal score machine only memorizes

We first discuss why any diffusion model that learns the ideal score function on a finite dataset can only memorize.

The key idea behind diffusion models is to reverse a stochastic forward diffusion process that iteratively converts the data distribution $\pi_0(\phi)$, where $\phi \in \mathbb{R}^N$ is any data point, into a sequence of distributions $\pi_t(\phi)$ over time $t$, such that the final distribution $\pi_T(\phi)$ at time $T$ is an isotropic Gaussian $\mathcal{N}(0, I)$. The forward diffusion process usually shrinks the data points toward the origin while adding Gaussian noise, so that when conditioning on any *individual* data point $\varphi \sim \pi_0$, the conditional probability $\pi_t(\phi|\varphi)$ becomes the Gaussian $\mathcal{N}(\phi|\sqrt{\bar{\alpha}_t}\varphi, (1 - \bar{\alpha}_t)I)$. The noise schedule $\bar{\alpha}_t$ decreases from 1 at $t = 0$ to 0 at $t = T$ so that the mean $\sqrt{\bar{\alpha}_t}\varphi$ of $\pi_t(\phi|\varphi)$ shrinks over time, and its variance increases, until $\pi_t(\phi|\varphi) \sim \mathcal{N}(0, I)$ for all initial points $\varphi$ (see figure 8).

A simple time reversal of this forward process can be obtained by sampling $\phi_T \sim \mathcal{N}(0, I)$ and then flowing it backwards in time from $T$ to 0 under the deterministic flow

$$-\dot{\phi}_t = \gamma_t(\phi_t + s_t(\phi_t)), \quad (1)$$

where $s_t(\phi) \equiv \nabla_\phi \log \pi_t(\phi)$ is the *score function* of the distribution $\pi_t(\phi)$ under the forward process and $\gamma_t$ depends on the entire noise schedule $\bar{\alpha}_t$ (see App. A for details). The flow in (1) induces a sequence of reverse distributions $\pi_t^R(\phi)$ that exactly reverse the forward process in the sense that

$\pi_t^R(\phi) = \pi_t(\phi)$ for all $t \in [0, T]$. Intuitively, this reversal occurs because, for any finite dataset $\mathcal{D}$, $\pi_t(\phi)$ is a mixture of Gaussians centered at shrunken data points,

$$\pi_t(\phi) = \frac{1}{|\mathcal{D}|} \sum_{\varphi \in \mathcal{D}} \mathcal{N}(\phi|\sqrt{\bar{\alpha}_t}\varphi, (1 - \bar{\alpha}_t)I), \quad (2)$$

and the score $s_t(\phi)$ points uphill on this mixture. Thus the second term in (1) flows $\phi_t$, as $t$ decreases, towards shrunken data points, and the first term undoes the shrinking.

Motivated by this theory, score-based diffusion models attempt to sample the data distribution $\pi_0(\phi)$ by forming an estimate $\hat{s}_t(\phi)$ of the score function $s_t(\phi)$, and then plugging this estimate and initial noise $\phi_T \sim \mathcal{N}(0, I)$ into the reverse flow in (1) to obtain a sample $\phi_0$. We consider what happens when the estimate matches the ideal score function so $\hat{s}_t(\phi) = s_t(\phi)$ on any finite dataset $\mathcal{D}$. Then the score of the Gaussian mixture $\pi_t(\phi)$ in (2), is (App. A):

$$s_t(\phi) = \frac{1}{1 - \bar{\alpha}_t} \sum_{\varphi \in \mathcal{D}} (\sqrt{\bar{\alpha}_t}\varphi - \phi) W_t(\varphi|\phi), \quad (3)$$

$$W_t(\varphi|\phi) = \frac{\mathcal{N}(\phi|\sqrt{\bar{\alpha}_t}\varphi, (1 - \bar{\alpha}_t)I)}{\sum_{\varphi' \in \mathcal{D}} \mathcal{N}(\phi|\sqrt{\bar{\alpha}_t}\varphi', (1 - \bar{\alpha}_t)I)}. \quad (4)$$

When $s_t$ in (3) is inserted into (1), each term in (3) acts as a force that pulls the sample $\phi$ towards a shrunken data point $\sqrt{\bar{\alpha}_t}\varphi$ as $t$ decreases, weighted by the posterior probability $W_t(\varphi|\phi)$ that $\phi$ at time $t$ would have originated from the datapoint $\varphi$ at time 0 under the forward diffusion.

The combined reverse dynamics in (1), (3) and (4), which we call the *ideal score machine*, has an appealing Bayesian guessing game interpretation: the current sample $\phi$ at time $t$ optimally guesses which data point $\varphi$ it originated from in the *forward process*, thereby forming the posterior belief distribution $W_t(\varphi|\phi)$, and then flows to each (shrunken version) of the data points, weighted by this belief.

Importantly, since the reverse flow provably reverses the forward diffusion, $\pi_0^R$ equals the empirical data distribution $\pi_0$, which is a sum of delta functions on the training set. Thus, *the ideal score machine memorizes.* The mechanism behind memorization can be explained by positive feedback instabilities in the reverse flow. In particular, the closer the sample $\phi$ is to a shrunken version of a data point $\varphi$, the higher the belief $W_t(\varphi|\phi)$ that $\phi$ originated from $\varphi$, and the stronger the force term $(\sqrt{\bar{\alpha}_t}\varphi - \phi)W_t(\varphi|\phi)$ in (3) pulling $\phi$ even closer to the shrunken $\varphi$, which in turn raises the belief $W_t(\varphi|\phi)$ at earlier $t$. This positive feedback between belief and force causes the posterior belief distribution $W_t(\varphi|\phi)$ to rapidly concentrate onto a *single* data point $\varphi$, and so $\phi_t$ flows to this same point $\varphi$ under the reverse flow (Fig.2 a.).

Thus, any diffusion model that learns the true score $s_t$ on a finite dataset $\mathcal{D}$ *must* memorize the training data and *cannot*

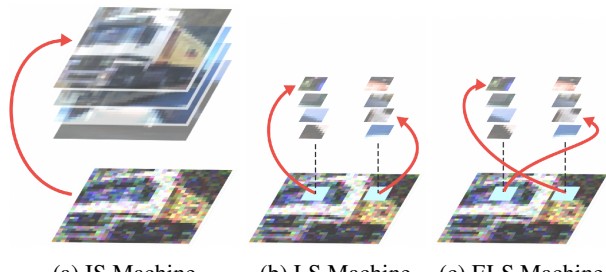

(a) IS Machine    (b) LS Machine    (c) ELS Machine

*Figure 2.* Ideal score-matching under various constraints. (a) In the IS machine, the *entire* image (bottom) reverse flows to a *single* training set image from the training set (top stack). (b,c) In both the LS and ELS machines, different local patches of the image flow to different local patches in the training set. In the LS machine this final training patch must be drawn from the *same* location (b), while in the ELS machine, it can be drawn from *any* location (c).

creatively generate new samples *far* from the training data. While we have explained this memorization phenomenon intuitively using the ideal score machine, it has been well established in prior work (e.g. Biroli et al. (2024)).

## 3. Equivariant and local score machines

The failure of creativity in the ideal score machine means that it *cannot* be a good model of what realistic diffusion models do beyond the memorization regime. We therefore seek simple inductive biases that *prevent* learning the ideal score function $s_t$ in (3) on a finite dataset $\mathcal{D}$. By identifying these inductive biases, we hope to obtain a new theory of what diffusion models do when they creatively generate new samples far from the training data.

The key observation is that many diffusion models use convolutional neural networks (CNNs) to form an estimate $\hat{s}(\phi)$ of the score function. Such CNNs have two prominent inductive biases. The first is translational equivariance due to weight sharing: translating the input image will correspondingly translate the CNN outputs. More generally, networks can be equivariant to arbitrary symmetry groups (e.g. (Cohen & Welling, 2016), (Hoogeboom et al., 2022)). The second is locality: since convolutional filters have narrow support, typical outputs of a CNN depend on their inputs only through a small receptive field of neighboring pixels. We therefore seek an optimal estimate $\hat{s}(\phi)$ of the ideal score in (3) subject to locality and equivariance constraints.

We start with formal definitions of equivariance and locality. Let $M_t[\phi]$ denote a model score function that takes an input image $\phi$ and outputs an estimated score $\hat{s}_t(\phi) = M_t[\phi]$.

**Definition 3.1.** A model $M_t$ is defined to be $G$-equivariant with respect to the action of a group $G$ on data if for any $U \in G$, $M_t$ satisfies $M_t[U\phi] = UM_t[\phi]$.

In our case of images, $G$ is the spatial translation group in two dimensions, $U\phi$ is a translated image, and $UM_t[\phi]$ is the translated score function. In other words, translating the input translates the outputs of an equivariant model in the same way. CNNs are translation equivariant if we impose periodic boundary conditions on the pixels, so that, for example, left translation of the leftmost pixels move them to the rightmost pixels (i.e. circular padding). However, the common practice of zero-padding images at their boundary breaks translation-equivariance; we extend our theory to this case in Sec. 3.4.

We next turn to locality. For image data, let $x$ be a pixel location, $\phi(x) \in \mathbb{R}^C$ be the pixel value of image $\phi$ at location $x$ (where $C$ is the number of color channels) and let $M_t[\phi](x) \in \mathbb{R}^C$ denote the model score function evaluated at pixel location $x$, which informs how the pixel value $\phi(x)$ should move under the reverse flow. Also, at each pixel location $x$, let $\Omega_x$ denote a local neighborhood of $x$ consisting of a subset of pixels near $x$, and let $\phi_{\Omega_x} \in \mathbb{R}^{|\Omega_x| \times C}$ be the restriction of pixel values of the entire image $\phi$ to the $|\Omega_x|$ pixels in the neighborhood $\Omega_x$. We define locality as:

**Definition 3.2.** $M_t[\phi]$ is defined to be $\Omega$-local if, for all images $\phi$ and all pixel locations $x$, $M_t[\phi](x)$ depends on $\phi$ only through $\phi_{\Omega_x}$, i.e. $M_t[\phi](x) = M_t[\phi_{\Omega_x}](x)$.

Thus if an $\Omega$-local model $M_t[\phi]$ is used in place of $s(t)$ in (1), the instantaneous reverse flow of any pixel value $\phi(x)$ at location $x$ and time $t$ will *not* depend on pixel values at any locations *outside* the local neighborhood $\Omega_x$; it depends *only* on the image *in* neighborhood $\Omega_x$. In particular, two pixels at distant locations $x$ and $y$ with non-overlapping neighborhoods $\Omega_x$ and $\Omega_y$ will make completely independent decisions as to which directions to reverse flow; the portion of the image $\phi_{\Omega_y}$ in the neighborhood $\Omega_y$ of $y$, cannot instanteously affect the flow direction of the pixel value $\phi(x)$, and vice versa.

Next, we consider the optimal minimum mean squared error (MMSE) approximation to the ideal score function $s_t(\phi)$ in (3) under locality and/or equivariance constraints. We provide full derivations in App. B, but the final answers, which we state below, are simple and intuitive.

### 3.1. The equivariant score (ES) machine

We first impose equivariance without locality. The MMSE equivariant approximation to $s(t)$ in (3)-(4) is identical in form to the ideal score, except the dataset $\mathcal{D}$ is augmented to the orbit of $\mathcal{D}$ under the equivariance group $G$, which we denote by $G(\mathcal{D})$. For example, in our case of images, $G(\mathcal{D})$ corresponds to all possible spatial translations of all images in $\mathcal{D}$. Explicitly, the MMSE equivariant score is given by

(see App. B.3 for a proof)

$$M_t[\phi](x) = \frac{1}{1 - \bar{\alpha}_t} \sum_{\varphi \in G(\mathcal{D})} (\sqrt{\bar{\alpha}_t}\varphi(x) - \phi(x))W_t(\varphi|\phi) \tag{5}$$

$$W_t(\varphi|\phi) = \frac{\mathcal{N}(\phi|\sqrt{\bar{\alpha}_t}\varphi, (1 - \bar{\alpha}_t)I)}{\sum_{\varphi' \in G(\mathcal{D})} \mathcal{N}(\phi|\sqrt{\bar{\alpha}_t}\varphi', (1 - \bar{\alpha}_t)I)}. \tag{6}$$

Replacing the ideal score $s(t)$ in (1) with (5) yields the equivariant score (ES) machine. While the ideal score machine memorizes the training data (see Sec. 2), the ES machine on images achieves only limited creativity: it can only generate any translate of any training image.

### 3.2. The local score (LS) machine

We next impose locality without equivariance. The MMSE $\Omega$-local approximation to $s(t)$ in (3)-(4) is given by

$$M_t[\phi](x) = \sum_{\varphi \in \mathcal{D}} \frac{(\sqrt{\bar{\alpha}_t}\varphi(x) - \phi(x))}{1 - \bar{\alpha}_t} W_t(\varphi_{\Omega_x}|\phi_{\Omega_x}), \tag{7}$$

$$W_t(\varphi_{\Omega_x}|\phi_{\Omega_x}) = \frac{\mathcal{N}(\phi_{\Omega_x}|\sqrt{\bar{\alpha}_t}\varphi_{\Omega_x}, (1 - \bar{\alpha}_t)I)}{\sum_{\varphi' \in \mathcal{D}} \mathcal{N}(\phi_{\Omega_x}|\sqrt{\bar{\alpha}_t}\varphi'_{\Omega_x}, (1 - \bar{\alpha}_t)I)}. \tag{8}$$

Each term in the local $M_t[\phi](x)$ in (7) is identical to each term in $s(t)$ in (3), yielding a force pulling the pixel value $\phi(x)$ towards a shrunken training set pixel value $\sqrt{\bar{\alpha}_t}\varphi(x)$ as before, *except* for the important change that the global posterior belief $W_t(\varphi|\phi)$ in (3)-(4), that is the same for *all* pixels $x$, is now replaced with a local $x$-dependent belief $W_t(\varphi_{\Omega_x}|\phi_{\Omega_x})$ in (7)-(8). $W_t(\varphi_{\Omega_x}|\phi_{\Omega_x})$ is the posterior probability that a sample image $\phi$ under the forward process at time $t$ originated from a training image $\varphi$ at time 0, conditioned on the only information the model $M_t[\phi](x)$ can depend on, namely the restriction $\phi_{\Omega_x}$ of the image $\phi$ to the local neighborhood $\Omega_x$ at location $x$. The closer the local image patch $\phi_{\Omega_x}$ is to the co-located training image patch $\varphi_{\Omega_x}$, the larger the posterior $W_t(\varphi_{\Omega_x}|\phi_{\Omega_x})$ in (8).

Replacing the ideal score $s(t)$ in (1) with (7) yields the local score (LS) machine. The LS machine can achieve significant combinatorial creativity by allowing local image neighborhoods $\phi_{\Omega_x}$ and $\phi_{\Omega_{x'}}$ of different pixels $x$ and $x'$ to reverse flow close to training image patches $\varphi_{\Omega_x}$ and $\varphi'_{\Omega_{x'}}$ from *different* training images $\varphi$ and $\varphi'$ (Fig.2b). Indeed the same positive feedback between belief and force that holds for the IS machine at a global level (Sec. 2), also holds for the LS machine at a local level, causing the posterior beliefs $W_t(\varphi|\phi_{\Omega_x})$ of all pixels $x$ to concentrate on a unique training image, but this training image could be different for different far away pixels. This flow decoupling of local image patches in $\phi_t$ empowers exponential creativity.

However, an important limitation remains in the LS machine: a local image patch $\phi_{\Omega_x}$ at pixel location $x$ *must* reverse flow close to some local training image patch $\varphi_{\Omega_x}$ drawn from the *same* location $x$; it cannot flow to a training image patch $\varphi_{\Omega_{x'}}$ drawn from a *different* location $x'$. We next see that adding equivariance removes this limitation.

### 3.3. The equivariant local score (ELS) machine

Further constraining the LS machine with equivariance leads to the ELS machine in which any local image patch at any pixel location $x$ can now flow towards any local training set image patch drawn from *any* location $x'$ not necessarily equal to $x$, as in the LS machine. This is the local analog of how the IS machine can only generate training set images, but the equivariance constrained ES machine can generate training set images globally translated to any other location.

To formally express this result, assume all local neighborhoods $\Omega_x$ for different $x$ have the same shape $\Omega$. For concreteness, one can think of $\Omega$ as a $P \times P$ square patch of pixels for $P$ odd, with $\Omega_x$ centered at location $x$. Then let $P_\Omega(\mathcal{D})$ denote the set of all possible $\Omega$ shaped local training image patches drawn from any training image centered at any location. An element $\varphi \in P_\Omega(\mathcal{D})$ now lives in $\mathbb{R}^{P \times P \times C}$ and denotes the pixel values of some local $\Omega$-shaped training image patch centered at some location. Now the optimal MMSE approximation to the ideal score in (3), under *both* equivariance and locality constraints is (App. B):

$$M_t[\phi](x) = \sum_{\varphi \in P_\Omega(\mathcal{D})} \frac{(\sqrt{\bar{\alpha}_t}\varphi(0) - \phi(x))}{1 - \bar{\alpha}_t} W_t(\varphi|\phi, x) \tag{9}$$

$$W_t(\varphi|\phi, x) = \frac{\mathcal{N}(\phi_{\Omega_x}|\sqrt{\bar{\alpha}_t}\varphi, (1 - \bar{\alpha}_t)I)}{\sum_{\varphi' \in P_\Omega(\mathcal{D})} \mathcal{N}(\phi_{\Omega_x}|\sqrt{\bar{\alpha}_t}\varphi', (1 - \bar{\alpha}_t)I)}. \tag{10}$$

We note that (9)-(10) for the ELS machine is identical to (7)-(8) for the LS machine except that: (1) the sum over local training set patches in (9)-(10) in determining the flow $M_t[\phi](x)$ for pixel $\phi(x)$ is no longer restricted to training patches centered at the same location as $x$; and (2) each pixel $x$ must now track a larger posterior belief state $W_t(\varphi|\phi, x)$ in (10) about which local training set patch at *any* location $x'$ was the origin of $\phi_{\Omega_x}$, as opposed to the smaller belief state $W_t(\varphi_{\Omega_x}|\phi_{\Omega_x})$ in (8) about which local training set patch at the *same* location $x$ was the origin of $\phi_{\Omega_x}$. In essence, in the Bayesian guessing game interpretation, equivariance removes each pixel's knowledge of its location $x$, so to guess the origin of its local image patch $\phi_{\Omega_x}$, it must guess both the training image *and* the location in the training image that it came from under the forward process. This guess then informs the reverse flow. Taken together, the ELS machine can creatively generate exponentially many novel images by

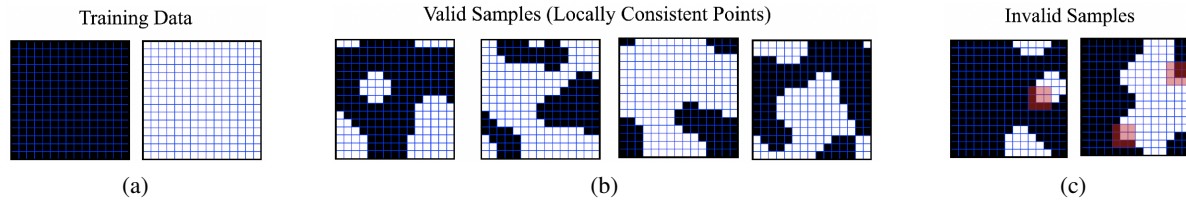

*Figure 3.* Exponential creativity through locally consistent patch mosaics. (a) A training set of two images (all black or all white). (b) Original samples from any local score machine (LS or ELS) with a $3 \times 3$ locality window and periodic boundary conditions. Local consistency in this special case means every generated pixel is either black or white, and the majority color of every generated $3 \times 3$ patch equals the color of its central pixel. (c) We note that samples are generated by numerically integrating the reverse flow in (1). If the step size in this integration is too large, one can generate invalid samples with a few cases of broken local consistency (highlighted red patches). In practice in trained diffusion models, this local consistency would only hold approximately.

mixing and matching local training set patches and placing them at any location in the generated image. We call this a *patch mosaic model of creativity.*

### 3.4. Breaking equivariance through boundaries

Due to the common practice of zero padding images at boundaries, CNNs actually break exact translational equivariance. We can modify our ELS machine to handle this broken equivariance (see App. B.2 for details). The key idea is that breaking translation equivariance restores to each pixel some knowledge of its location within the image. For example, if the local image patch $\phi_{\Omega_x}$ around pixel location $x$ contains many 0 values, then the pixel can use these to infer its location with respect to the boundary, and use this knowledge in the Bayesian guessing game that determines the reverse flow. In essence, with additional conditioning about its relation to the boundary, $\phi_{\Omega_x}$ should *only* flow to training image patches that are consistent with the observed amount and location of zero-padding. For example, interior, edge, and corner image patches only flow to interior, edge and corner training image patches with the same boundary overlap (Fig. 9). This is a partial case of complete equivariance breaking in the LS machine, in which pixels know their exact location $x$, and the local image patch $\phi_{\Omega_x}$ only flows to training image patches at the *same* location $x$ (Fig.2b).

## 4. A theory of creativity after convergence

It is clear that the reverse flow from Gaussian noise $\phi_T$ to final sample $\phi_0$ in the ideal score machine converges to a single training set image. But what do the LS, ELS or boundary broken ELS machines converge to at the end of the reverse process if they creatively generate novel samples *far* from the training data? We answer this question by proving a theorem that characterizes the converged samples $\phi = \phi_0$ at the end of the reverse process (App. B.4).

**Theorem 4.1.** *For the LS, ELS, and boundary broken ELS machines, assuming $\lim_{t \to 0} \phi_t$ and $\lim_{t \to 0} \partial_t \phi_t$ exist, then for every pixel $x$, $\phi_0(x) = \varphi(0)$ for the unique patch $\varphi \in$*

$P_\Omega^x(\mathcal{D})$ *for which $\phi_{\Omega_x}$ is closer in $L_2$ distance (in $\mathbb{R}^{|\Omega_x| \times C}$) than other local training set patch $\varphi' \in P_\Omega^x(\mathcal{D})$.*

Intuitively, samples generated from these machines are *locally consistent* in the sense that they obeying 3 local conditions: (1) every pixel $x$ can be uniquely assigned to a local training set patch $\varphi$; (2) the pixel value $\phi_0(x)$ is *exactly* equal to the central pixel $\varphi(0)$ of $\varphi$; (3) the rest of the local generated patch $\phi_{\Omega_x}$ resembles the local training patch $\varphi$ more than any other possible training patch. This result characterizes the creative outcome of locally constrained machines as creating *locally consistent* patch mosaics where every pixel of every local patch in the sample matches the central pixel of the $L_2$ closest local patch in the training set.

### 4.1. The simplest example of patch mosaic creativity

As the simplest possible example illustrating the locally consistent patch mosaic model of creativity for the LS and ELS machines, consider a training set of *only* two images: an all black and an all white image (Fig.3a). A highly expressive diffusion model trained only on these two images would only generate these two images. However, an LS or ELS machine with local $3 \times 3$ neighborhoods generates exponentially many new samples that are locally consistent patch mosaics (Fig.3b): every pixel is either black or white, indicating it is assigned to either an all black or all white $3 \times 3$ local training set patch. And any $3 \times 3$ local patch of a generated sample with a central black (white) pixel is closer to the all black (white) training set patch than the other training set patch. Thus local consistency in this special case reduces to the simple condition that the majority color of any $3 \times 3$ locally generated patch must equal the color of its central pixel. The reader can check that this local consistency holds (with appropriate circular wraparound) at every pixel in Fig.3b.

## 5. Tests of the theory on trained models

We next test our theory on two CNN-based architectures, a standard UNet (Ronneberger et al., 2015) and a ResNet

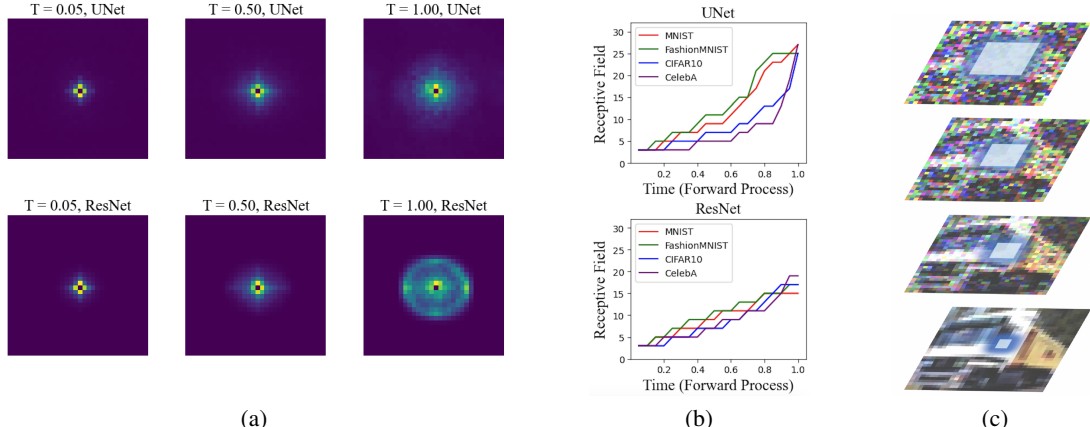

*Figure 4.* Coarse to fine progression of spatial locality in the reverse flow. (a) A heatmap of the average absolute value of the Jacobian from the output score $M_t[\phi_t](x = 0)$ at the center pixel $x = 0$ back to all input pixels $\phi(x')$ as a function of $x'$. This receptive field shrinks from large to small as time progresses from early (large $t$) to late (small $t$) in the reverse flow. (b) Optimally calibrated values of the spatial locality scale $P$ of the (E)LS machine as a function of time $t$ (see App. C.2 for details of calibration). (c) A schematic view of the time-dependent LS and ELS machines in which the locality neighborhood shrinks as the reverse time flows from top to bottom.

(He et al., 2016) trained on 4 datasets, MNIST, Fashion-MNIST, CIFAR10, and CelebA (see App. C.1 for details of architectures and training). We restrict our attention to these simple datasets because our theory is for CNN-based diffusion models only, and more complex diffusion models with attention and latent spaces are required to model more complex datasets.

### 5.1. Coarse-to-fine time dependent spatial locality scales

To compare our theory of ELS and LS machines with experiments, we must first choose a locality scale for the size of the $P \times P$ local patch. We measure it in the trained UNet and ResNet and find, importantly, that it changes from large to small scales as time passes from early (large $t$) to late (small $t$) in the reverse flow (Fig. 4a). We therefore promote the spatial size of the $P \times P$ locality window in our ELS and LS machines to a dynamic variable which we calibrate to the UNet and ResNet (Fig. 4bc). See App. C.2.

### 5.2. Theory predicts trained outputs case-by-case

We first compare the outputs of the scale-calibrated boundary broken-ELS machine to the outputs of the ResNet and the UNet on a case-by-case basis for the same initial noise samples $\phi_T$ to both the theory and the ResNet or UNet, and we find an excellent match (Fig. 5ab). Indeed we find **a remarkable and uniform *quantitative* agreement between the CNN outputs and ELS machine outputs.** For ResNets, we find median $r^2$ values between theory and experiment of 0.94 on MNIST, 0.90 on FashionMNIST, 0.90 on CIFAR10, and 0.96 on CelebA32x32. For UNets, we find median $r^2$ values of 0.89 on MNIST, 0.93 on FashionMNIST, and 0.90 on CIFAR10 (see Fig. 10 for the full distribution of $r^2$ values). We find, unlike on other datasets, that the UNet

behavior is more accurately described by the *local score machine* rather than the ELS machine on CelebA32x32, the former achieving median $r^2 \sim 0.90$; we describe this observation in more detail in section 5.4. To our knowledge, this is *the first time* an analytic theory has explained the creative outputs of a trained deep neural network-based generative model to this level of accuracy. Importantly, the (E)LS machine explains all trained outputs far better than the IS machine (Fig. 10 and Table 2). See App. D, Fig. 13 to Fig. 22 for many more successful case-by-case theory-experiment comparisons for the 2 nets and 3 datasets.

We also trained circularly padded ResNets on MNIST and CIFAR10, and found a good match between the *non-boundary broken ELS machine* and experiment (Figs. 11, 21 and 22). Interestingly, in both theory and experiment for MNIST, circular padding yields more texture-like outputs and less localized digit-like outputs, indicating the fundamental importance of boundaries in anchoring diffusion models, for MNIST at least (compare Fig. 21 and Fig. 13).

### 5.3. Spatial inconsistencies from excess late-time locality

Diffusion models notoriously generate spatially inconsistent images at fine spatial scales, e.g. incorrect numbers of fingers and limbs. Indeed, these inconsistencies are considered a tell-tale sign of AI-generated images (Bird & Lotfi, 2024; Shen et al., 2024; Lin et al., 2024). Our trained models on FashionMNIST also generate such inconsistencies, e.g. pants with too many or too few legs, shoes with more than one toe region, and shirts with incorrect numbers of arms. Remarkably, our theory, since it matches trained model outputs on a case by case basis, *also* reproduces these inconsistencies (Fig. 5c). Since our theory is completely mechanistically interpretable, it provides a clear explanation

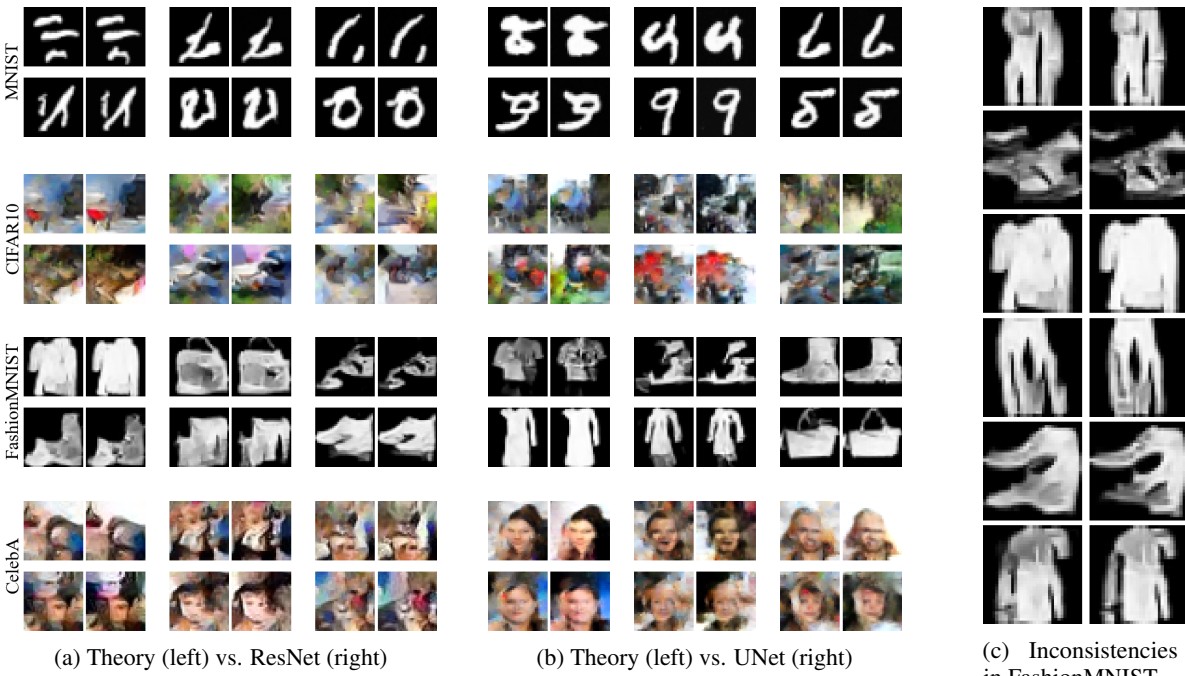

(a) Theory (left) vs. ResNet (right)  (b) Theory (left) vs. UNet (right)  (c) Inconsistencies in FashionMNIST

*Figure 5.* Match between theory and experiment. (a,b) Each pair of images shows a striking match between the output of the boundary broken ELS machine (left image in each pair) and the output of a trained CNN diffusion model (right image in each pair) when both models are given the same initial noise input. We compare theory with 2 architectures (ResNet in (a), and UNet in (b)) on 3 datasets (MNIST, CIFAR10 and FashionMNIST from top to bottom). See App. D, Fig. 13 to Fig. 22 for many comparisons and Fig. 10 and Table 2 for quantitative $r^2$ values indicating high match between theory and experiment. (c) Trained CNN diffusion models (right) produce well-known spatial inconsistencies (e.g. 3 legged pants (row 1,4), 3 armed tops (row 3,6), bifurcated shoes (row 2,5)). Remarkably, the ELS theory (left) predicts this behavior and mechanistically explains it through excessive spatial locality at late times in the reverse flow.

for the origin of these inconsistencies in terms of excessive locality at late stages of the reverse flow. The late-time ($t < 0.3$) locality for all models is less than about 5 pixels (Fig. 4b). With such a small locality scale, different parts of the image more than a few pixels away must decide whether to develop into e.g. an arm or a pant leg without knowing the total number of limbs in the image; this process frequently results in incorrect numbers of total limbs.

### 5.4. UNets can fully break equivariance

We note that for three datasets, MNIST, FashionMNIST and CIFAR10, the best matching theory that explains the outputs of zero-padded CNNs (for both ResNets and UNets) is the boundary-broken ELS machine (see Table 2 and Fig. 5).

However, interestingly, for CelebA, an LS machine that fully breaks equivariance better explains the outputs of the UNet, but not the ResNet, compared to the boundary broken ELS (Table 2). Indeed, the UNet creates more structured faces than the ResNet (compare rows 2 and 4 in figure Fig. 7). The less structured faces of the ResNet are better explained by the boundary-broken ELS machine (compare rows 1 and 2 in Fig. 7), while the more structured faces of the UNet are better explained by the LS machine with fully broken equivariance (compare rows 3 and 4 in Fig. 7).

An explanation for why the UNet can in principle fully break equivariance, while the ResNet cannot, is that the maximal possible receptive field (RF) size of the ResNet is 17x17 while the image is 32x32. Thus, at any instant of time $t$, the

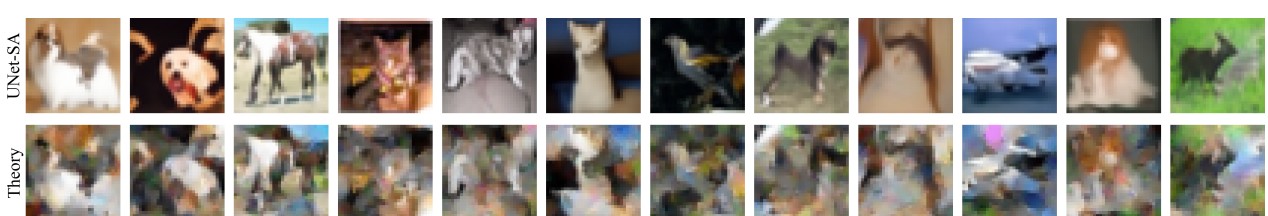

*Figure 6.* Comparison between UNet+SA outputs (top row) and ELS machine outputs (bottom row) for the same noise inputs. For this class of inputs, the UNet+SA appears to carve out more semantically coherent objects out of the closely related ELS patch mosaic.

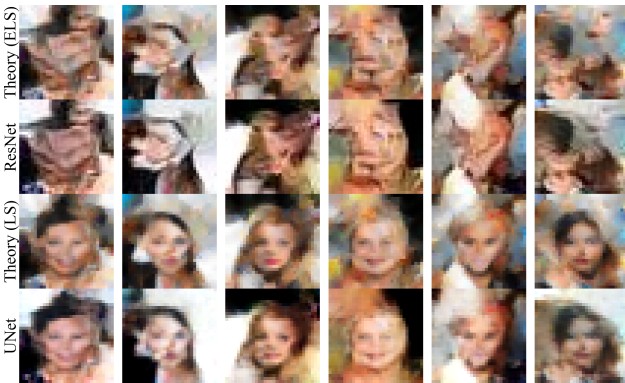

*Figure 7.* Comparison between LS, ELS, ResNet, and UNet outputs on CelebA. Each column represents identical initial noise inputs. Note the Unet produces more spatially structured faces than the ResNet, and the LS machine better explains this higher UNet performance than the ELS machine.

ResNet score computation at pixels near the image center cannot depend on image data outside this RF. However, the maximal possible RF size of the UNet covers the entire image. Thus, the UNet can in principle use information over the entire image, including the boundary, to infer the absolute location of each pixel when computing the score at that pixel. Indeed, it does this for CelebA, possibly because for CelebA there are strong correlations between image neighborhoods and pixel locations (e.g. eyes, ears, mouths and noses all appear in similar locations across the dataset). However, for the other datasets, the UNet does not seem to infer absolute pixel location far from the boundary when computing the score at each instant of time, and so is better described by a boundary-broken ELS machine rather than an LS machine with fully broken equivariance.

## 6. The relation between theory and attention

While the local theory explains the outputs of CNN-based diffusion models on a case by case basis with high accuracy, many diffusion models also include highly non-local self-attention (SA) layers. For example (Ho et al., 2020)) added SA layers to a UNet (which we call a UNet-SA architecture). The non-locality of SA strongly violates the assumptions of our local theory. This violation raises an important question: do the predictions of our local theory bear *any resemblance at all* to the non-local outputs of trained UNet+SA models?

To address this question, we compare our existing ELS machine theory with the outputs of a publicly available UNet+SA model pretrained on CIFAR10 (Sehwag, 2024). Strikingly, our ELS model, with no modification whatsoever, predicts the UNet+SA outputs on a case-by-case basis with a median of $r^2 \sim 0.77$ on 100 sample images. This is substantially higher than the median $r^2 \sim 0.48$ of an IS machine baseline on the same images (see Fig. 12 for the

entire distribution of $r^2$ values).

Qualitatively, the outputs of the UNet+SA model fall into three rough classes in which the UNet+SA produces: (1) a semantically incoherent image which nevertheless strongly resembles the prediction of the ELS machine (Fig. 23a); (2) a semantically coherent image which has some quantitative correlation with, but little qualitative resemblance to, the ELS machine prediction (Fig. 23b); and (3) a semantically coherent image that *also* has a strong resemblance to the less semantically coherent ELS machine outputs (Fig. 6).

This third class is the largest and most interesting of the three. Qualitatively, the UNet+SA appears to carve a semantically coherent object out of the patch mosaic of the ELS machine (compare top and bottom rows of Fig. 6). For example, the UNet+SA often cuts out a foreground object from the ELS patch mosaic, while smoothing the background and accentuating it from the foreground object.

Fig. 24 shows a large set of comparisons between the ELS machine and UNet+SA outputs. While these results show that the ELS theory bears in many cases both quantitative and qualitative resemblance to the UNet+SA outputs, a full quantitative theory of the role of attention in the creativity of diffusion models remains for future investigation. However, the correspondences in Fig. 6, Fig. 23a, and Fig. 24 and the ELS correlations (y-axis) in Fig. 12, suggest the ELS theory provides an important foundation for this endeavor.

## 7. Discussion

Developing a mechanistic understanding of how generative models convert their training data into novel outputs *far* from their training data is an important goal in the field of neural network interpretability. We have developed such an understanding for convolutional diffusion models of images that accurately predicts *individual* outputs on fixed random inputs in terms of the training data, for standard architectures (ResNets and UNets), standard datasets (MNIST, FashionMNIST, CIFAR10, and CelebA), and standard loss functions (score-matching). Moreover, our mechanistically interpretable theory of diffusion models is derived not from intensive and highly detailed analysis of the inner workings of trained networks (modulo matching spatial scales), as in most mechanistic interpretability works, but rather from a first principles approach stemming from analytic solutions for the optimal score subject to *only* 2 posited inductive biases: locality and equivariance. The strong quantitative agreement between theory and experiment on a case-by-case basis suggests that these two inductive biases are *sufficient* to explain the creativity of convolution-only diffusion models. We hope this work provides a foundation for understanding the creativity of more powerful attention-enabled diffusion models trained on more complex datasets.

## Acknowledgements

M.K. would like to acknowledge the support of the NSF Graduate Research Fellowship. M.K. would like to acknowledge the helpful conversations, comments, and feedback from Daniel Kunin, Atsushi Yamamura, and Feng Chen. S.G. thanks the Simons Foundation, a Schmidt Sciences Polymath Award, and an NSF CAREER award for funding.

## Impact Statement

This paper presents work whose goal is to advance our understanding of Machine Learning systems. As the scope of the capabilities of these systems increase, and as these systems become more deeply integrated into socially important applications, it is imperative to develop a fundamental understanding of how these capabilities emerge. Unfortunately, the development of such fundamental understanding has lagged significantly behind advances in capabilities. Our work helps address this gap by developing a better understanding of simple but still highly nontrivial deep networks, hopefully paving the way for future studies that move our fundamental understanding of these systems closer to the state-of-the-art of capabilities.

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

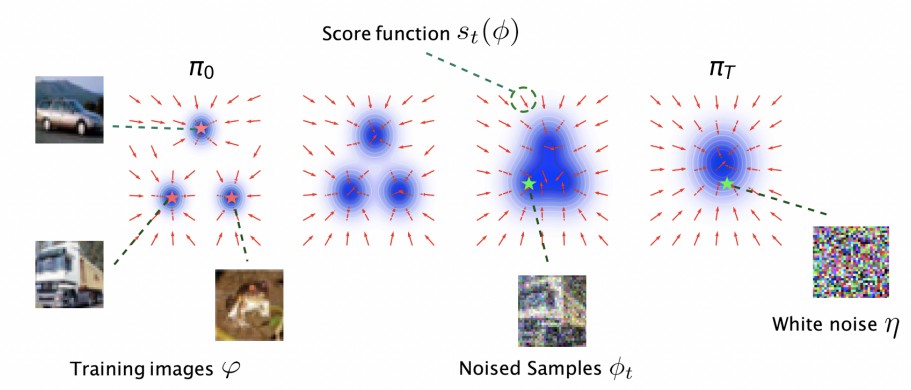

*Figure 8.* A schematic illustration of score-matching diffusion.

# A. Mathematical Preliminaries

## A.1. Notation conventions

In what follows, we use the following notation:

- $\mathcal{D}$ will represent the training set.

- $\varphi \in \mathbb{R}^N$ will represent an example from the training set. For images of size $L$ pixels by $L$ pixels by $C$ channels, we have $N = L \times L \times C$.

- $\phi$ will represent any arbitrary image (or other data) that we are plugging into the score function/diffusion model.

- $x$ represents a pixel location in an image.

- For image data, $\phi(x)$ and $\varphi(x)$ will represent the pixel values of the images $\phi$ and $\varphi$ at pixel location $x$; both are elements of $\mathbb{R}^C$.

- $M[\phi] : \mathbb{R}^N \to \mathbb{R}^N$ represents a model that takes in an image $\phi$ and produces a new image (e.g. an estimate of the score function). We will denote by $M[\phi](x) \in \mathbb{R}^C$ the value of the outputs of this model, given an input $\phi$, at the pixel location $x$.

- $\phi_{\Omega_x}$ and $\varphi_{\Omega_x}$ will represent the restriction of images $\phi$ and $\varphi$ to a neighborhood $\Omega_x$ around a pixel $x$. We usually take $\Omega_x$ to be a square patch of size $P \times P$, with $P$ odd, containing pixel $x$ at the center. In this case, $\phi_{\Omega_x}$ and $\varphi_{\Omega_x}$ are vectors in $\mathbb{R}^{P \times P \times C}$. However, the theoretical framework supports arbitrary assignments from $x \to \Omega_x$.

- For a square image patch $\varphi$ with an odd-dimension side length, the value $\varphi(0) \in \mathbb{R}^C$ indicates the pixel at the center of the patch.

- $P_\Omega(\mathcal{D})$ will denote the set of all $\Omega$-shaped patches drawn from elements of $\mathcal{D}$.

- $\mathcal{N}(x | \mu, \Sigma)$ represents the PDF of the normal distribution with mean $\mu$ and covariance $\Sigma$. We also use the short-hand $\mathcal{N}(\mu, \Sigma)$ when we do not need to refer to the name of a specific random variable.

## A.2. Stochastic differential equations (SDEs) and Probability Flow

In probabilistic modeling, we are often confronted with the problem of sampling from a data distribution whose exact form we do not have access to, or whose form makes direct sampling difficult. Diffusion models are an approach to sampling from such distributions by learning a time-inhomogenous differential equation that transports samples from a simple Gaussian distribution to the more complex distribution of interest.

More formally, consider a time-dependent (Itô) stochastic differential equation, given as follows:

$$d\phi_t = f_t(\phi_t)\, dt + g_t\, dW_t. \tag{11}$$

Here $W_t$ is a standard Wiener process and $dW_t$ is its differential. We call this stochastic process the 'forward' process. It starts from the data distribution $\pi_0(\phi)$ and induces a flow on probability distributions $\pi_t(\phi)$ for $t \geq 0$ described by associated Fokker-Planck equation:

$$\frac{\partial \pi_t(\phi)}{\partial t} = -\nabla \cdot (f_t(\phi)\pi_t(\phi)) + \frac{1}{2}\nabla^2(g_t^2\pi_t(\phi)). \tag{12}$$

We will imagine that our forward process is constructed so that as $t \to \infty$ (or as $t \to T$ for some finite time $T$), $\pi_t$ converges to some tractable $\pi_\infty$, typically a Gaussian with finite variance.

The idea underpinning diffusion models (or, more technically, DDIMs, the deterministic variant of diffusion models considered for the most part in this paper) is to look for a *deterministic, time-dependent vector field* $v_t(\phi)$ that induces the same flow on distributions as (12). Then one can simply reverse this flow to sample from $\pi_0(t)$ by first sampling from the simple distribution $\phi_T \sim \pi_T$, then evolving the sample deterministically backwards in time from $t = T$ to $t = 0$ under the ODE

$$\frac{d\phi_t}{dt} = v_t(\phi_t). \tag{13}$$

This ODE induces a flow on probability distributions $\pi_t(\phi)$ described by the advection equation

$$\frac{\partial \pi_t}{\partial t} = -\nabla \cdot [v_t(\phi)\pi_t(\phi)]. \tag{14}$$

We want this advection process above to induce the *same flow* on distributions as the original flow (12), when run in reverse starting, from the simple final distribution $\pi_T$. (This setup is closely related to 'flow matching' models: see (Lipman et al., 2022) for a review). Interestingly, $v_t(\phi)$ can be easily identified by rewriting the flow in (12) as

$$\frac{\partial \pi_t(\phi)}{\partial t} = -\nabla \cdot ([f_t(\phi) - \frac{1}{2}g_t^2 \nabla \log \pi_t(\phi)]\pi_t(\phi)). \tag{15}$$

By matching (14) and (15), we find

$$v_t(\phi) = f_t(\phi) - \frac{1}{2}g_t^2 \nabla \log \pi_t(\phi). \tag{16}$$

This vector field is sometimes known as the 'probability flow.' The function

$$s_t(\phi) = \nabla \log \pi_t(\phi) \tag{17}$$

is known as the *score function*, and contains all of the complicated dependency on the initial distribution $\pi_0(\phi)$ that we would like to capture in our model.

### A.3. Diffusion models

The most common choice of forward process (11) is an inhomogenous Ornstein–Uhlenbeck (OU) process process of the following form:

$$d\phi_t = -\gamma_t\phi_t + \sqrt{2\gamma_t}dW_t \tag{18}$$

for which the probability flow is given by

$$v_t(\phi) = -\gamma_t(\phi + \nabla \log \pi_t(\phi)). \tag{19}$$

The reason for this choice is that the finite-time marginals $\pi_t$ for this distribution can be sampled from tractably. We can generate samples $\phi_t \sim \pi_t$ by computing the following linear linear combination:

$$\phi_t = \sqrt{\bar{\alpha}_t}\phi_0 + \sqrt{1 - \bar{\alpha}_t}\eta_t \tag{20}$$

with $\phi_0 \sim \pi_0$ a sample from the target distribution and $\eta_t \sim \mathcal{N}(0, I)$ a vector of isotropic Gaussian noise. The values of $\bar{\alpha}_t$ depend on the choice of $\gamma_t$ via the following formula:

$$\bar{\alpha}_t = \exp\left(-2 \int_0^t \gamma_t \, dt\right). \tag{21}$$

In practice, the values $\bar{\alpha}_t$ are typically chosen first and $\gamma_t$ is then specified implicitly by this choice. The choice of $\bar{\alpha}_t$ is known as the 'noise schedule' for a diffusion model; typically, we choose $\bar{\alpha}_0 = 1$ (so that $t = 0$ corresponds to uncorrupted sample) and $\bar{\alpha}_T = 0$ for some large but finite value of $T$ (so that the entire reverse process can take place in finite time). At a distributional level, the solution of (12) for this process is given by

$$\pi_t(\phi) = \int \pi_0(\phi_0) \mathcal{N}(\phi | \sqrt{\bar{\alpha}_t} \phi_0, (1 - \bar{\alpha}_t) I) \, d\phi_0. \tag{22}$$

The score function for $\pi_t$ can then be obtained analytically in terms of $\pi_0$:

$$
\begin{aligned}
s_t(\phi) &= -\frac{1}{1 - \bar{\alpha}_t} \int \frac{\pi_0(\phi_0) \mathcal{N}(\phi | \sqrt{\bar{\alpha}_t} \phi_0, (1 - \bar{\alpha}_t) I)}{\pi_t(\phi)} (\phi_t - \sqrt{\bar{\alpha}_t} \phi_0) \, d\phi_0 \\
&= -\frac{1}{1 - \bar{\alpha}_t} \int \mathbb{P}(\phi_0 | \phi_t = \phi)(\phi_t - \sqrt{\bar{\alpha}_t} \phi_0) \, d\phi_0.
\end{aligned}
\tag{23}
$$

There is an extremely convenient fact about this particular score function that we can take advantage of in order to learn it from data. Given a particular sample $\phi_t$ generated by the forward noising process, the score function is proportional to the conditional expectation of the added noise $\eta_t$ from (20), given $\phi_t$:

$$s_t(\phi) = -\frac{1}{\sqrt{1 - \bar{\alpha}_t}} \langle \eta_t | \phi_t = \phi \rangle. \tag{24}$$

This result is known as Tweedie's theorem. A standard result in statistics is that the conditional expectation $\langle \eta_t | \phi_t \rangle$ is the functional optimum of the following loss function:

$$\mathcal{L}_t(f) = \mathbb{E}_{\phi_0 \sim \pi_0, \eta_t \sim \mathcal{N}(0, I)}[\| f(\phi_t(\phi_0, \eta_t)) - \eta_t \|^2] \tag{25}$$

for $\phi_t$ defined in (20); the following slightly rescaled loss can be used if score-matching is preferred:

$$\mathcal{L}_t(f) = \mathbb{E}_{\phi_0 \sim \pi_0, \eta_t \sim \mathcal{N}(0, I)}[\| f(\phi_t(\phi_0, \eta_t)) + (1 - \bar{\alpha}_t)^{-1/2} \eta_t \|^2]. \tag{26}$$

In practice, we model the score using a single neural network $f_\theta(x, t)$ for all times $t \in [0, T]$, using the following objective:

$$L(\theta) = \mathbb{E}_{t \sim U(0, T), \phi_0 \sim \pi_0, \eta_t \sim \mathcal{N}(0, I)}[\| f_\theta(\phi_t(\phi_0, \eta_t), t) - \eta_t \|^2]. \tag{27}$$

### A.4. The empirical score function

In practice, we never have direct access to the data distribution $\pi_0$ that we are attempting to sample from; we only have access to the discrete empirical prior defined by a particular training set $\mathcal{D}$:

$$\pi_0(\phi) = \frac{1}{|\mathcal{D}|} \sum_{\varphi \in \mathcal{D}} \delta(\phi - \varphi). \tag{28}$$

At finite time $t$, the empirical distribution of noised training examples is simply a mixture of Gaussians centered at the (rescaled) training data points:

$$\pi_t(\phi) = \frac{1}{|\mathcal{D}|} \sum_{\varphi \in \mathcal{D}} \mathcal{N}(\phi | \sqrt{\bar{\alpha}_t} \varphi, (1 - \bar{\alpha}_t) I). \tag{29}$$

The score function (23) for this distribution is then simply given by

$$s_t(\phi) = -\frac{1}{1 - \bar{\alpha}_t} \sum_{\varphi \in \mathcal{D}} (\phi - \sqrt{\bar{\alpha}_t}\varphi) W_t(\varphi|\phi), \tag{30}$$

$$W_t(\varphi|\phi) = \frac{\mathcal{N}(\phi|\sqrt{\bar{\alpha}_t}\varphi, (1 - \bar{\alpha}_t)I)}{\sum_{\varphi' \in \mathcal{D}} \mathcal{N}(\phi|\sqrt{\bar{\alpha}_t}\varphi', (1 - \bar{\alpha}_t)I)}. \tag{31}$$

Intuitively, this corresponds to computing the conditional average over the added noise, by averaging the proposed noise vectors $\eta_t \propto (\phi - \sqrt{\bar{\alpha}_t}\varphi)$ between our observed example $\phi$ and each training example $\varphi$, weighted by the probability $W(\varphi|\phi)$ of $\varphi$ being the training example that $\phi$ originated from. This probability is in turn computed essentially by Bayes theorem: the probability of starting from a training example $\varphi$, given the observed $\phi$, is given by the likelihood of generating the noise needed to go from $\varphi$ to $\phi$, divided by the likelihood of going from $\varphi'$ to $\phi$ for all possible training examples $\varphi'$. Appealingly, the weights $W(\varphi|\phi)$ are given by computing a simple soft-max over a simple quadratic loss function $-\frac{1}{2(1-\bar{\alpha}_t)}\|\phi - \sqrt{\bar{\alpha}_t}\varphi\|^2$ for every point in the training set.

It should be emphasized at this point that the ideal score function is *not* representative of real diffusion models. Primarily: it always memorizes the training data. More importantly in practice, this memorization property becomes manifest *very early* in the reverse process for high dimensional data, due to the typically large separation between training points in Euclidean space. This is a manifestation of the curse of dimensionality– it would require an amount of data *exponential in the dimension* to provide sufficiently good coverage of the underlying space for the ideal *empirical* score function to well-approximate the *true* ideal score function over all inputs over all times.

The failure of the ideal score function as a model for realistic diffusion models suggests that we should try to understand the particular manner in which they fail to optimally solve the task that they are trained on. In particular, we are motivated to look for the *implicit and explicit biases and constraints* that prevent these models from learning the ideal score function, and then understand what they do instead under these limitations.

## B. Formalism

### B.1. Optimal local translationally equivariant score matching

Fully translationally equivariant local models $M_t$ can be written in the following way:

$$M_t[\phi](x) = f[\phi_{\Omega_x}], \tag{32}$$

where $\phi_{\Omega_x}$ is the restriction of $\phi$ to the neighborhood $\Omega_x$ around pixel $x$. In this section, we will use circular boundary conditions, so that if $x$ is near an image border, the neighborhood $\phi_{\Omega_x}$ includes the pixels on the opposite side of the image near the corresponding border (we will revisit this in the next section). This functional form reflects the locality constraint by making manifest that the output at a pixel location $x$ depends only on the patch $\phi_{\Omega_x}$ around it. Equivariance is reflected in the fact that the output of the model at every point $x$ is determined by the same function of the input patch. $f$ should be thought of as a function mapping $\mathbb{R}^{C \times |\Omega|} \to \mathbb{R}^C$, where $C$ is the number of channels in the image and $|\Omega|$ is the number of pixels in the local patch $\Omega$. The problem of identifying the optimal local/equivariant model can thus be framed as finding the $f$ that minimizes the score matching objective:

$$\mathcal{L} = \sum_x \mathbb{E}_{\phi \sim \pi_t}[\|f[\phi_{\Omega_x}] - s_t[\phi](x)\|^2] \tag{33}$$

Writing this out concretely gives

$$\mathcal{L} = \int \pi_t(\phi) \sum_x \|f(\phi_{\Omega_x}) - s_t[\phi](x)\|^2 \, d\phi. \tag{34}$$

To find the functional optimum, we vary the objective with respect to $f(\Phi)$, with $\Phi$ any arbitrary patch, and set this variation to zero. This yields the condition

$$0 = \sum_x \int \pi_t(\phi)(f(\phi_{\Omega_x}) - s_t[\phi](x))\delta(\phi_{\Omega_x} - \Phi) \, d\phi. \tag{35}$$

We can rearrange this into the following form:

$$f(\Phi) \sum_x \pi_t(\phi_{\Omega_x} = \Phi) = \sum_x \int \delta(\phi_{\Omega_x} - \Phi)\, \pi_t(\phi) s_t[\phi](x)\, d\phi$$

$$= \sum_x \int \delta(\phi_{\Omega_x} - \Phi) \nabla_{\phi(x)} \pi_t(\phi)\, d\phi$$

$$= \sum_x \nabla_{\Phi(0)} \pi_t(\phi_{\Omega_x} = \Phi)$$

Here $\Phi(0) \in \mathbb{R}^C$ is the pixel value in the center of the patch $\Phi$. $\pi_t(\phi_{\Omega_x} = \Phi)$ indicates the marginal probability under the distribution $\pi_t$ that the patch $\phi_{\Omega_x}$ equals the target patch $\Phi$. The distribution $\sum_x \pi_t(\phi_{\Omega_x} = \Phi)$ is then proportional to the marginal distribution that a randomly-selected $\Omega$-shaped-patch in the image $\phi$ equals $\Phi$. Dividing through by this marginal, we obtain

$$f(\Phi) = \nabla_{\Phi(0)} \log \sum_x \pi_t(\phi_{\Omega_x} = \Phi) \tag{36}$$

i.e. we find that $f(\Phi)$ is simply the score function of the modified marginal density $\sum_x \pi_t(\phi_{\Omega_x} = \Phi)$. Since $\pi_t(\phi)$ is a mixture of Gaussians, the marginal $\pi_t(\phi_{\Omega_x} = \Phi)$ can be obtained simply and is given by

$$\pi_t(\phi_{\Omega_x} = \Phi) = \sum_{\varphi \in \mathcal{D}} \mathcal{N}(\Phi | \sqrt{\bar{\alpha}_t} \varphi_{\Omega_x}, (1 - \bar{\alpha}_t)I). \tag{37}$$

Summing over $x$ gives us

$$\sum_x \pi_t(\phi_{\Omega_x} = \Phi) = \sum_{\varphi \in P_\Omega(\mathcal{D})} \mathcal{N}(\Phi | \sqrt{\bar{\alpha}_t} \varphi, (1 - \bar{\alpha}_t)I) \tag{38}$$

where $P_\Omega(\mathcal{D})$ is the set of all $\Omega$ patches in the training set $\mathcal{D}$. Finally, taking the derivative with respect to $\Phi(0)$ and substituting $\phi_{\Omega_x}$ for $\Phi$ gives us the final answer for $f[\phi_{\Omega_x}]$, which, when inserted into (32), yields the final answer for $M_t$:

$$M_t[\phi](x) = -\frac{1}{1 - \bar{\alpha}_t} \sum_{\varphi \in P_\Omega(\mathcal{D})} (\phi(x) - \sqrt{\bar{\alpha}_t}\varphi(0)) W(\varphi | \phi_{\Omega_x}) \tag{39}$$

$$W(\varphi | \phi_{\Omega_x}) = \frac{\mathcal{N}(\phi_{\Omega_x} | \sqrt{\bar{\alpha}_t}\varphi, (1 - \bar{\alpha}_t)I)}{\sum_{\varphi' \in P_\Omega(\mathcal{D})} \mathcal{N}(\phi_{\Omega_x} | \sqrt{\bar{\alpha}_t}\varphi', (1 - \bar{\alpha}_t)I)}. \tag{40}$$

We term the reverse diffusion model parameterized by $M_t$ i (39) the Equivariant Local Score (ELS) Machine.

This result has a simple intuitive interpretation. Firstly, it should be noted that the form of the resulting approximation to the score function strongly resembles the form of the true score function (30). In that case, the score function computation could be framed as guessing the added noise by finding the necessary added noise for each possible training set element, computing the likelihood of generating that noise under a Gaussian noise model, and then averaging the possible noises over the entire training set weighted by the Bayesian posterior over each possible noised example.

The ELS machine (39) can be interpreted similarly. However, a very important distinction is that the *Bayes weights are pixelwise-decoupled.* Under the exact computation of the score function, the Bayes weights are computed based on all available information in the image, and shared across every pixel; under the locality-constrained approximation, each pixel independently computes a separate set of Bayes weights for each training set element, based on its local receptive field. This decoupling of the belief states of different pixels means that under the reverse denoising process parameterized by (39), *different pixels will be drawn towards different elements of the training set*. At scales below the locality scale the final denoised images should (roughly) resemble part of a training set image; however, at larger scales, the resulting images will not resemble any particular training set image, but rather a kind of patchwork quilt/mosaic of randomly combined training set images. We make this result more precise in (B.4).

The role played by equivariance can likewise be interpreted very simply as removing each pixel's ability to locate itself within the image. Position is therefore promoted to a latent variable that must be integrated over, in addition to the training set element itself. This results in needing to compute a Bayes weight not only for each correspondingly-located patch in the training set, but *every possible patch* in the training set.

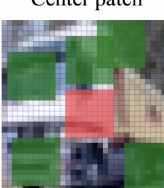

*Figure 9.* In the presence of zero-padded borders, different dictionaries of training set patches are used for the ELS machine computation depending on the contextual information provided by the visible border within the patch. For a central patch (left, red patch) without border information, training set patches (green patches) are sourced from the entire image interior. For edge patches (middle, red patch), training set patches (green patches) are sourced from everywhere along the edge at the same distance from the border. For corner patches (right, red patch), only patches from that exact location are used in the computation.

## B.2. Adding borders

There is an ambiguity about the behavior of a convolutional neural network for pixels near enough to the boundary of an image such that the network's receptive field extends past that boundary. One option in that situation is to enforce circular boundary conditions, so that the convolution operation 'wraps around' to the other side upon encountering the boundary. This approach is not typically used in practice; more commonly, 'zero padding' is introduced, wherein pixels outside of the image are treated as zeros for the purposes of the convolution operation.

In the presence of zero-padding, the results given above concerning the optimal local equivariant approximation to the score are nearly identical; in fact, the fundamental identity (32) still holds. However, we must modify the interpretation of the visible patch $\phi_{\Omega_x}$ for a pixel $x$ near the border. Instead of considering the patch to include 'wrapped around' portions of the image, we instead simply extend it with zeros in all locations where it extends past the border.

When the ELS machine takes as input the patch $\phi_{\Omega_x}$, it computes the conditional probability that it corresponds to a noising of each particular patch in the training set. Formally, getting an exactly zero value at any pixel location occurs with probability zero. Thus, observing a patch $\phi_{\Omega_x}$ with zero-padding indicates with probability 1 that the patch is a corruption of a training set patch that came from a location inside the image consistent with the observed border information. We are thus able to write the ELS machine in the presence of a zero-padded boundary as

$$M_t[\phi](x) = -\frac{1}{1-\bar{\alpha}_t} \sum_{\varphi \in P_\Omega^x(\mathcal{D})} (\phi(x) - \sqrt{\bar{\alpha}_t}\varphi(0)) \frac{\mathcal{N}(\phi_{\Omega_x}|\sqrt{\bar{\alpha}_t}\varphi, (1-\bar{\alpha}_t)I)}{\sum_{\varphi' \in P_\Omega^x(\mathcal{D})} \mathcal{N}(\phi_{\Omega_x}|\sqrt{\bar{\alpha}_t}\varphi', (1-\bar{\alpha}_t)I)}. \tag{41}$$

The only modification to the ELS machine (39) is that we have replaced the set of all patches $P_\Omega(\mathcal{D})$ in the sum with the $x$-dependent patch dictionary $P_\Omega^x(\mathcal{D})$, corresponding to the collection of patches consistent with the border data at location $x$. These collections are illustrated in figure 9.

## B.3. Optimal equivariant score matching for a general symmetry group

In many diffusion model applications outside of computer vision, equivariance under more general symmetry groups is built in to the architecture of the backbone model. For instance, molecular diffusion models are sometimes made equivariant under $E(3)$, the group of isometries on Euclidean space (Hoogeboom et al., 2022). Diffusion transformers (Peebles & Xie, 2023) are also naturally equivariant under the group of sequence permutations, although this equivariance is broken in a controlled way by the inclusion of positional embeddings. We are thus motivated to study the question of optimality under the constraint of equivariance under a general group of symmetries $G$, which we define as follows:

**Definition B.1.** Let $G$ be a particular group of transformations acting on data $\phi$. We say that a model $M_t$ is $G$-equivariant if, for any $U \in G$, our model satisfies

$$M_t[U\phi] = U M_t[\phi]. \tag{42}$$

The result is given here:

**Theorem B.2.** *The optimal $G$-equivariant approximation to the empirical score function (3) under the score matching objective (26) is given by the empirical score function for the dataset $G(\mathcal{D})$ consisting of the orbit of the dataset $\mathcal{D}$ under the group $G$.*

*Proof.* Let $M_t$ be a $G$-equivariant model. For simplicity, we will assume that $M_t$ is being optimized with the following loss:

$$L_t = \mathbb{E}_{\phi \sim \pi_t}[\|M_t[\phi] - s_t(\phi)\|^2] \tag{43}$$

where $s_t = \nabla_\phi \log \pi_t(\phi)$ is the ideal score function. First consider the orbit of a single point $\phi_0$ under the group $G$, given by $G[\phi_0] = \{\phi : \exists U \in G : U\phi_0 = \phi\}$. For any $\phi \in G[\phi_0]$, there is an element $U \in G$ such that $U^{-1}\phi = \phi_0$, and thus the output of an equivariant model $M_t[\phi]$ is simply $U M_t[\phi_0]$. The problem of picking an optimal $M_t[\phi]$ for any $\phi \in G[\phi_0]$ can thus be reduced to a standard linear regression for $M_t[\phi_0]$, under the loss

$$\begin{aligned}
\tilde{L}_t &= \mathbb{E}_{\phi \sim \pi_t | \phi \in G(\phi_0)}[\|M_t[\phi] - \nabla \log \pi_t(\phi)\|^2] \\
&= \int_G \frac{\pi_t[U^{-1}\phi_0]}{\pi_t(G[\phi_0])} \|M_t[\phi] - U\nabla \log \pi_t(U^{-1}\phi_0)\|^2 \, dU
\end{aligned}$$

where in the second line we have used the property of unitaries that $\|Ux\|^2 = \|x\|^2$. Here $\pi_t(G[\phi_0])$ indicates the probability density assigned to the entire orbit $G[\phi_0]$ by $\pi_t$. We have used the orbit-stabilizer property to write the integral over the orbit as an integral over the entire group. Despite its complexity this formula represents a standard least-squares objective for $M_t[\phi]$, the minimizer of which is simply the weighted average of the target function $U\nabla \log \pi_t(U^{-1}\phi_0)$ weighted by $\frac{\pi_t[U^{-1}\phi_0]}{\pi_t(G[\phi_0])}$. In other words, our optimal $G$-equivariant model is

$$M_t[\phi] = \int_{U \in G} U \, \nabla \log \pi_t[U^{-1}\phi] \frac{\pi_t(U^{-1}\phi)}{\int_{V \in G} \pi_t(V^{-1}\phi) \, dV} dU. \tag{44}$$

We can do some simple algebra to write this experssion in a more interpretable form:

$$\frac{\int_{U \in G} U \, \nabla \log \pi_t[U^{-1}\phi]\pi_t(U^{-1}\phi) \, dU}{\int_{U \in G} \pi_t(U^{-1}\phi) \, dU} = \frac{\int_{U \in G} U\nabla \pi_t[U^{-1}\phi] \, dU}{\int_{U \in G} \pi_t(U^{-1}\phi) \, dU} = \nabla_\phi \log \int_{U \in G} \pi_t[U^{-1}\phi] \, dU$$

where in the last step we have used the fact that $U^{-1} = U^\dagger$ and that $\nabla_\phi f(U^\dagger \phi) = U[\nabla f](U^\dagger \phi)$. We now note that

$$\int_{U \in G} \pi_t[U^{-1}\phi] \, dU = \frac{1}{|\mathcal{D}|} \sum_{\varphi \in \mathcal{D}} \int_{U \in G} \mathcal{N}(U^{-1}\phi; \varphi\sqrt{\bar{\alpha}_t}, (1 - \bar{\alpha}_t)I) dU. \tag{45}$$

Since $U$ is unitary, it follows that

$$\begin{aligned}
\mathcal{N}(U^{-1}\phi|\varphi\sqrt{\bar{\alpha}_t}, (1 - \bar{\alpha}_t)I) &\propto \exp\left(-\frac{\|U^{-1}\phi - \sqrt{\bar{\alpha}_t}\varphi\|^2}{2(1 - \bar{\alpha}_t)}\right) \\
&= \exp\left(-\frac{\|\phi - \sqrt{\bar{\alpha}_t}U\varphi\|^2}{2(1 - \bar{\alpha}_t)}\right)
\end{aligned}$$

and thus our optimal model is the score function for the empirical noise distribution of the $G$-augmented dataset, i.e.

$$\pi_t^G(\phi) = \frac{1}{|\mathcal{D}|} \sum_{\varphi \in \mathcal{D}} \int_{G(\varphi)} \mathcal{N}(\phi; \sqrt{\bar{\alpha}_t}\varphi', (1 - \bar{\alpha}_t)I) \, d\varphi' \tag{46}$$

$$M_t[\phi] = \nabla \log \pi_t^G(\phi) = -\frac{1}{1 - \bar{\alpha}_t} \frac{\sum_{\varphi \in \mathcal{D}} \int_{G(\varphi)}(\phi - \sqrt{\bar{\alpha}_t}\varphi')\mathcal{N}(\phi|\sqrt{\bar{\alpha}_t}\varphi', (1 - \bar{\alpha}_t)I) \, d\varphi'}{\sum_{\varphi \in \mathcal{D}} \int_{G(\varphi)} \mathcal{N}(\phi|\sqrt{\bar{\alpha}_t}\varphi', (1 - \bar{\alpha}_t)I) \, d\varphi'} \tag{47}$$

$\square$

**B.4. The sample distribution at $t = 0$ under a local score approximation**

When the score is learned optimally, the reverse process concentrates the sample distribution on the training dataset as $t \to 0$. It is instructive for us to ask what the analogous constraint on the generated samples is for the locality-constrained models that we consider in this paper. The answer is that the flow will concentrate the probability on certain 'locally consistent points' $\tilde{\phi}$, defined as follows. Suppose we are employing an $\Omega$-local approximation $M_t$ to the score function, with each individual pixel $x$ using a (possibly identical) dictionary of patches $\varphi \in P_\Omega^x$. A 'locally consistent point' $\tilde{\phi}$ is a point such that for every pixel location $x$, the value $\tilde{\phi}(x)$ is equal to the center pixel $\varphi(0)$ of the $l_2$-closest patch $\varphi \in P_\Omega^x$ to the patch $\tilde{\phi}_{\Omega_x}$, i.e. the patch that minimizes $\left\| \varphi - \tilde{\phi}_{\Omega_x} \right\|^2$ over all patches in $P_\Omega^x$.

The reverse flow approximation parameterized by $M_t$ will concentrate on locally consistent points. We can formalize this effect in the following theorem:

**Theorem B.3.** *Suppose we sample an initial point $\phi_T$ from the Gaussian $\pi_T$, and we evolve this density under the standard reverse process*

$$\partial_t \phi_t = -\gamma_t (\phi_t + M_t(\phi_t)) \tag{48}$$

*where*

$$\gamma_t = -\frac{\partial_t \bar{\alpha}_t}{2\bar{\alpha}_t}. \tag{49}$$

*Suppose also that the limits $\lim_{t \to 0} \phi_t$ and $\lim_{t \to 0} \partial_t \phi_t$ exist for an initial point $\phi_T$. Then the limit must be a locally consistent point.*

*Proof.* The assumption that $\lim_{t \to 0} \partial_t \phi_t$ exists entails that for any point $\phi_t$ on a particular trajectory, the values of $\phi_t$ and $-\gamma_t(\phi_t + M_t(\phi_t))$ must stay bounded as $t \to 0$, which in turn entails that $\gamma_t M_t(\phi_t)$ must likewise stay bounded as $t \to 0$. This latter quantity is given at pixel location $x$ by

$$\lim_{t \to 0} \gamma_t M_t[\phi](x) = \lim_{t \to 0} -\frac{\partial_t \bar{\alpha}_t}{2\bar{\alpha}_t(1 - \bar{\alpha}_t)} \sum_{\varphi \in P_\Omega^x} (\phi_t(x) - \sqrt{\bar{\alpha}_t}\varphi(0)) W(\varphi | \phi, x) \tag{50}$$

$$W(\varphi | \phi, x) = \frac{\mathcal{N}(\phi_{\Omega_x} | \sqrt{\bar{\alpha}_t}\varphi, (1 - \bar{\alpha}_t)I)}{\sum_{\varphi' \in P_\Omega^x} \mathcal{N}(\phi_{\Omega_x} | \sqrt{\bar{\alpha}_t}\varphi', (1 - \bar{\alpha}_t)I)}. \tag{51}$$

The prefactor goes to $\infty$ as $t^{-1}$ as $t \to 0$, so it follows that for the derivative to have a finite limit, the right-hand factor must go to zero. As $\bar{\alpha}_t \to 0$, the weights take the limiting values

$$\lim_{t \to 0} W(\varphi | \phi, x) = \begin{cases} 1 & \varphi = \arg\min_{\varphi' \in P_\Omega^x} \{\|\phi_{\Omega_x} - \varphi'\|^2\} \\ 0 & \text{else} \end{cases} \tag{52}$$

and thus the limiting value of the sum is simply

$$(\tilde{\phi}(x) - \tilde{\varphi}(0)) \tag{53}$$

where $\tilde{\varphi} = \arg\min_{\varphi' \in P_\Omega^x} \{\|\phi_{\Omega_x} - \varphi'\|^2\}$. This value is zero only when $\tilde{\phi}(x) = \tilde{\varphi}(0)$. The condition that this holds for all $x$ is the definition of a locally consistent point. $\square$

## C. Empirics

### C.1. Experimental details

To test our ELS machine model of CNN-based diffusion, we examine two different architectures:

1. UNet: we use a standard UNet (Ronneberger et al., 2015) with three scales with channel dimensions of $64, 128, 256$ respectively. We use residual connections in each UNet block. This model is formally local, but has a maximum receptive field size larger than the $32 \times 32$ images we consider.

2. ResNet: we use a minimal 8-layer convolutional neural network, with an upscaling and downscaling layer and 6 intermediate convolutional layers at a channel dimension of 256. Each layer is a single convolutional layer with a kernel size of $3 \times 3$ and with residual connections (He et al., 2016) between layers. This model has a formal maximum receptive field size of $17 \times 17$.

For all experiments, we train each model for 300 epochs with Adam, using an initial learning rate of 1e-4, a batchsize of 128, and an exponential learning rate schedule that applies a multiplicative factor of 0.999965 to the learning rate with each step (this approximately halves the learning rate over the course of 50 epochs with our batch size of 128).[1] We do not employ normalization layers in any of our models in order to avoid the possibility of information being exchanged nonlocally throughout the image. We do not use weight decay. We sample images from the theoretical processes (IS/LS/ELS) using a 20-step discretization of the reverse process, using the analytic form prescribed for DDIM-style models in (Song et al., 2020a). We sample from the CNN models using 20-step and 150-step discretizations, and report the respective results in tables 2 and **??**. As evidenced, we find that the 150-step discretization outputs better match the theoretical outputs than the 20-step discretization; no theoretical explanation for this has been suggested. We use a cosine noise schedule (Nichol & Dhariwal, 2021) for each experiment.

We evaluate each architecture using zero-padded convolutions on the following datasets: MNIST, FashionMNIST, CIFAR10, and CelebA. For each of the latter datasets, we use class-conditioning. We only train class-unconditional models on MNIST and CelebA. In addition, we evaluate our ResNet architecture using *circularly-padded convolutions* on CIFAR10 (class-conditional) and MNIST (class-unconditional).

For each neural network on each dataset, we calibrate an associated multiscale LS model and an associated multiscale ELS model of the network using the procedure described in C.2. The (E)LS model inherits the class-conditionality of the neural network it is modeling. We then compute the outputs of each neural network on 100 distinct random noise inputs for each dataset, drawn iid from an isotropic normalized Gaussian distribution. For class-conditional models, we additionally sample a label for each seed. We compute the outputs of the corresponding ELS machine on the same seeds/labels. For each example, we compute the pixelwise $r^2$ between the (E)LS machine outputs and the network output. We also compute the outputs of the IS machine across all of the same inputs/labels, and compute the pixelwise $r^2$ between this baseline and the neural network outputs. For all except the CelebA UNet we find that the ELS machine is the best predictor; for this case, we find that the LS machine is significantly more predictive, indicating that the model is able to use positional information in the interior of the image. We report the median $r^2$ value across the 100 samples for all configurations in table 2, and plot the distribution of ELS correlations/IS correlations in figures 10 and 11 (LS/IS correlations for the CelebA UNet).

We also repeat this analysis procedure for a pretrained self-attention-enabled UNet trained on CIFAR10 from (Sehwag, 2024), with the exception that we re-use the scales calibrated for the CIFAR10 ResNet model rather than re-estimating them for the Self-Attention-enabled model in order to minimize bias.

## C.2. Identifying multiscale behavior

In order to correctly recapitulate the behavior of the models we study, we need to account for a crucial empirical observation: *convolutional diffusion models exhibit time-dependent effective receptive field sizes*. This behavior is illustrated in figure 4. In the left panel, we display an average absolute value of the gradient of the center pixel of the model outputs from two of our CIFAR10-trained diffusion models, with respect to the input image. We plot this at various time steps in the reverse process (with the center pixel omitted for visual clarity). This visualization highlights which areas of the image the center pixel's outputs are sensitive to, an indicator of the degree of locality in the model's output. At $T = 1.0$ (corresponding to an input of initial white noise), the average gradient spans a large range (for the ResNet, a range clearly constrained by its maximal receptive field size). As the noise level reduces throughout the reverse process, the width of the heatmap decreases, until at the last time step the heatmap is almost entirely concentrated in a single ($3 \times 3$) square.

To calibrate the time-dependent locality scale of our theoretical model, we compute the reverse trajectories under the CNN-parameterized neural networks for a random validation set. At each time step, we compare the predictions of the model for the added noise and the outputs of ELS machines with a range of scales via cosine similarity. We then pick the representative scale for each time step by picking the median optimal scale across the range of samples. The resulting calibrated scales are shown in the middle panel of figure 4. We see that the UNet is better described by a larger-scale ELS machine early in the reverse process than its ResNet counterparts, a phenomenon that can probably be linked to the

---

[1]Code for the following experiments hosted at https://github.com/Kambm/convolutional_diffusion

more stringent locality constraints in the latter model. However, as the reverse process continues, both models prefer monotonically smaller scales, until converging to the smallest scale ($3 \times 3$) for the final few denoising steps. These results are in accordance with the visual evidence from the gradient heatmaps in figure 4.

At this stage we have no a-priori method for predicting the scales that the models choose to use at each time step. However, the general phenomenon where the model initially starts with a large field of view and decreases it over the course of the reverse process could be anticipated on general grounds. As the noise variance decreases, the noised training distribution separates into a multimodal distribution with a larger and larger number of modes; as $t \to 0$, the number of modes converges to the (very large!) number of training examples. Since the models we consider are equivariant, an optimal model must also in principle represent not just the modes corresponding to the training set, but also to every translated augmentation, which for a $32 \times 32$ image results in a 1024-fold increase in effective dataset size! However, the emergence of multimodality is delayed when considering only the marginal distributions with respect to a smaller scale, as there are fewer dimensions via which two distinct data points could be distinguished from each other. This suggests that the model may somehow be picking the largest scale that it can a) represent within its receptive field (a constraint more pertinent to the ResNet, which has a smaller maximum receptive field size) and b) for which it can represent the local/equivariant approximation to the score function in a *reasonably parameter efficient way*, i.e. for which it need not model too many independent modes of the data distribution. However, more work needs to be done in order to understand this phenomenon.

## D. Results

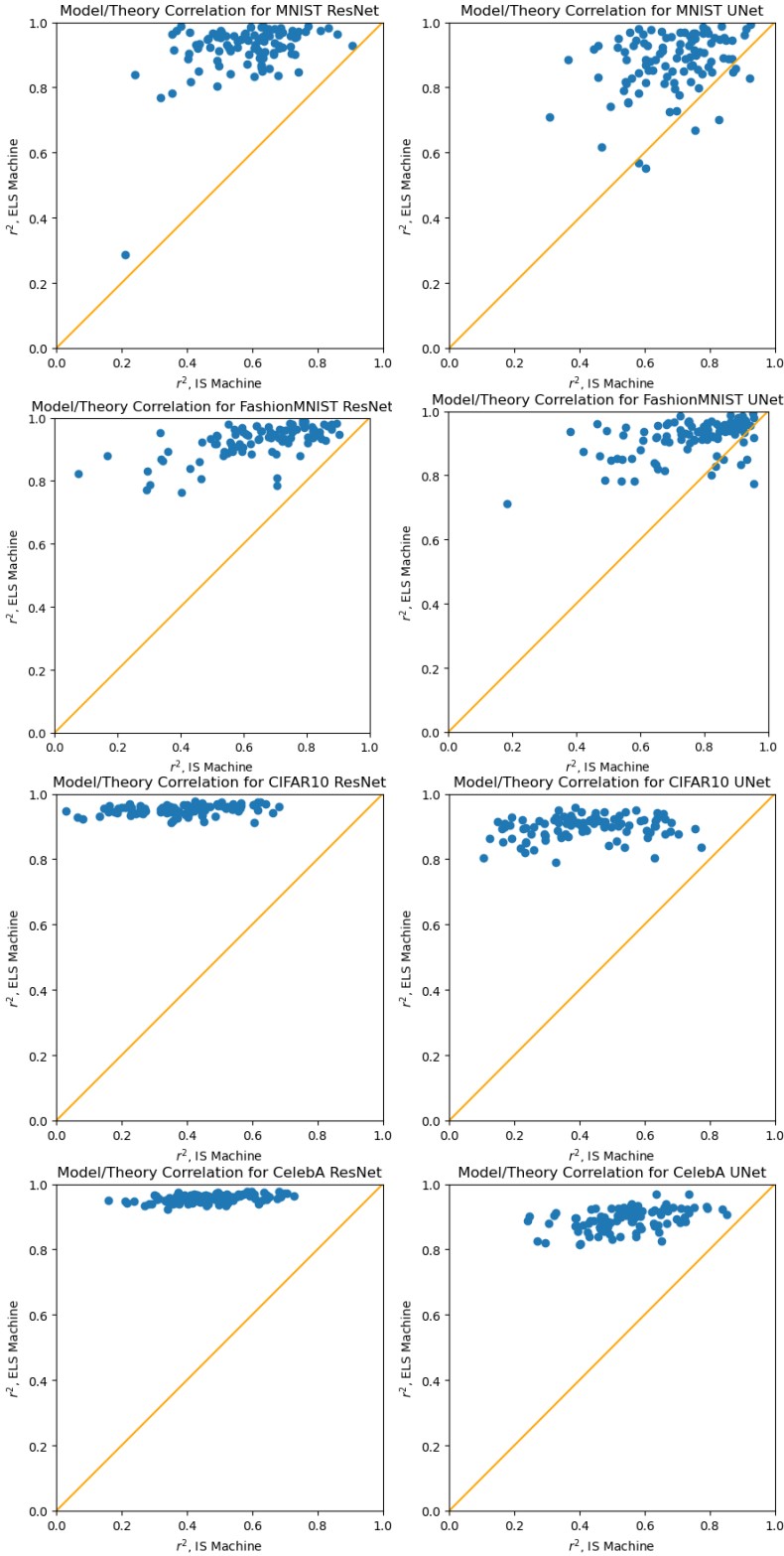

*Figure 10.* Correlations between model outputs and (E)LS machine/IS baseline on each dataset for zero-padded models. LS machine is used for the CelebA UNet, ELS machine for all other models. Y axis is (E)LS machine $r^2$, X axis is IS baseline $r^2$ for each data point in the sample. The (E)LS machine uniformly outperforms the memorizing baseline.

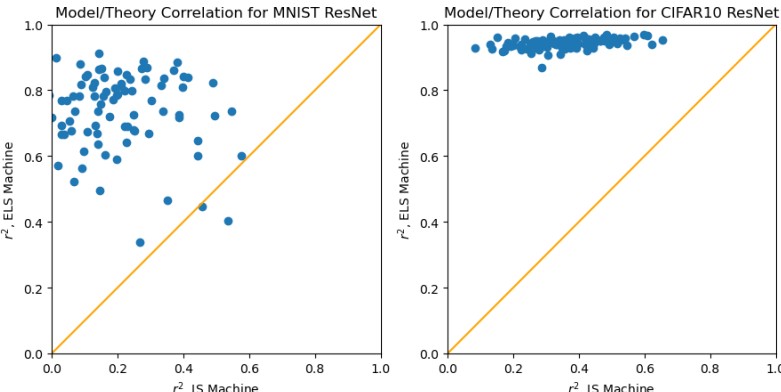

*Figure 11.* Correlations between model outputs and ELS machine/IS baseline on each dataset for circularly-padded models. Y axis is ELS machine $r^2$, X axis is IS baseline $r^2$ for each data point in the sample. The ELS machine uniformly outperforms the baseline. The performance of the ELS machine on circular MNIST is anomalously lower than other configurations, but the degree of outperformance of the ideal score baseline is higher.

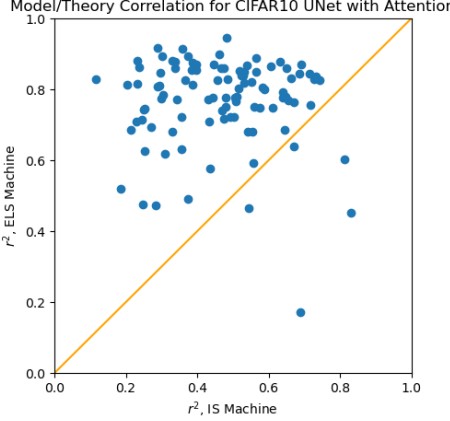

*Figure 12.* Correlations between model outputs and ELS machine/IS baseline on CIFAR10 for a pretrained Attention-enabled UNet. Y axis is ELS machine $r^2$, X axis is IS baseline $r^2$ for each data point in the sample.

*Table 1.* A summary of the experimental results of the paper for different datasets and model configurations for each architecture across each dataset, with 20 steps in the theoretical reverse process and 150 steps in the neural network reverse process. Pixelwise $r^2$ between theory and neural network images are computed using 100 image samples per configuration; the median across the sample is reported. We compare these results to a baseline consisting of the correlations of the model with the outputs of an ideal score (IS) model, which always outputs memorized training examples. We also report the percentage of samples on which the ELS machine outperforms the output from the ideal score-matching diffusion model.

| Dataset | Arch. | Padding | Conditional | ELS Corr. | LS Corr. | IS Corr. | (E)LS > IS % |
|---|---|---|---|---|---|---|---|
| MNIST | UNet | Zeros | ✗ | **0.89** | 0.88 | 0.70 | 0.93 |
| CIFAR10 | UNet | Zeros | ✓ | **0.90** | 0.87 | 0.41 | 0.92 |
| FashionMNIST | UNet | Zeros | ✓ | **0.93** | 0.93 | 0.80 | 1.00 |
| CelebA | UNet | Zeros | ✗ | 0.85 | **0.90** | 0.55 | 1.00 |
| MNIST | ResNet | Zeros | ✗ | **0.94** | 0.82 | 0.61 | 1.00 |
| MNIST | ResNet | Circular | ✗ | **0.77** | 0.36 | 0.15 | 0.92 |
| CIFAR10 | ResNet | Zeros | ✓ | **0.95** | 0.90 | 0.42 | 1.00 |
| CIFAR10 | ResNet | Circular | ✓ | **0.94** | 0.83 | 0.35 | 1.00 |
| FashionMNIST | ResNet | Zeros | ✓ | **0.94** | 0.88 | 0.68 | 1.00 |
| CelebA | ResNet | Zeros | ✗ | **0.96** | 0.90 | 0.47 | 1.00 |
| CIFAR10 | UNet + SA | Zeros | ✓ | **0.77** | 0.77 | 0.48 | 0.95 |

*Table 2.* A summary of the experimental results of the paper for different datasets and model configurations for each architecture across each dataset, with 20 steps in the theoretical reverse process and 20 steps in the neural network reverse process.

| Dataset | Arch. | Padding | Conditional | ELS Corr. | LS Corr. | IS Corr. | (E)LS > IS % |
|---|---|---|---|---|---|---|---|
| MNIST | UNet | Zeros | ✗ | **0.84** | 0.83 | 0.69 | 0.92 |
| CIFAR10 | UNet | Zeros | ✓ | **0.82** | 0.80 | 0.39 | 0.99 |
| FashionMNIST | UNet | Zeros | ✓ | **0.91** | 0.91 | 0.80 | 0.90 |
| CelebA | UNet | Zeros | ✗ | 0.75 | **0.81** | 0.51 | 1.00 |
| MNIST | ResNet | Zeros | ✗ | **0.94** | 0.84 | 0.62 | 0.97 |
| MNIST | ResNet | Circular | ✗ | **0.73** | 0.33 | 0.14 | 0.99 |
| CIFAR10 | ResNet | Zeros | ✓ | **0.90** | 0.86 | 0.43 | 1.00 |
| CIFAR10 | ResNet | Circular | ✓ | **0.90** | 0.80 | 0.36 | 1.00 |
| FashionMNIST | ResNet | Zeros | ✓ | **0.90** | 0.88 | 0.74 | 0.97 |
| CelebA | ResNet | Zeros | ✗ | **0.92** | 0.88 | 0.48 | 1.00 |
| CIFAR10 | UNet + SA | Zeros | ✓ | **0.75** | 0.74 | 0.47 | 0.92 |

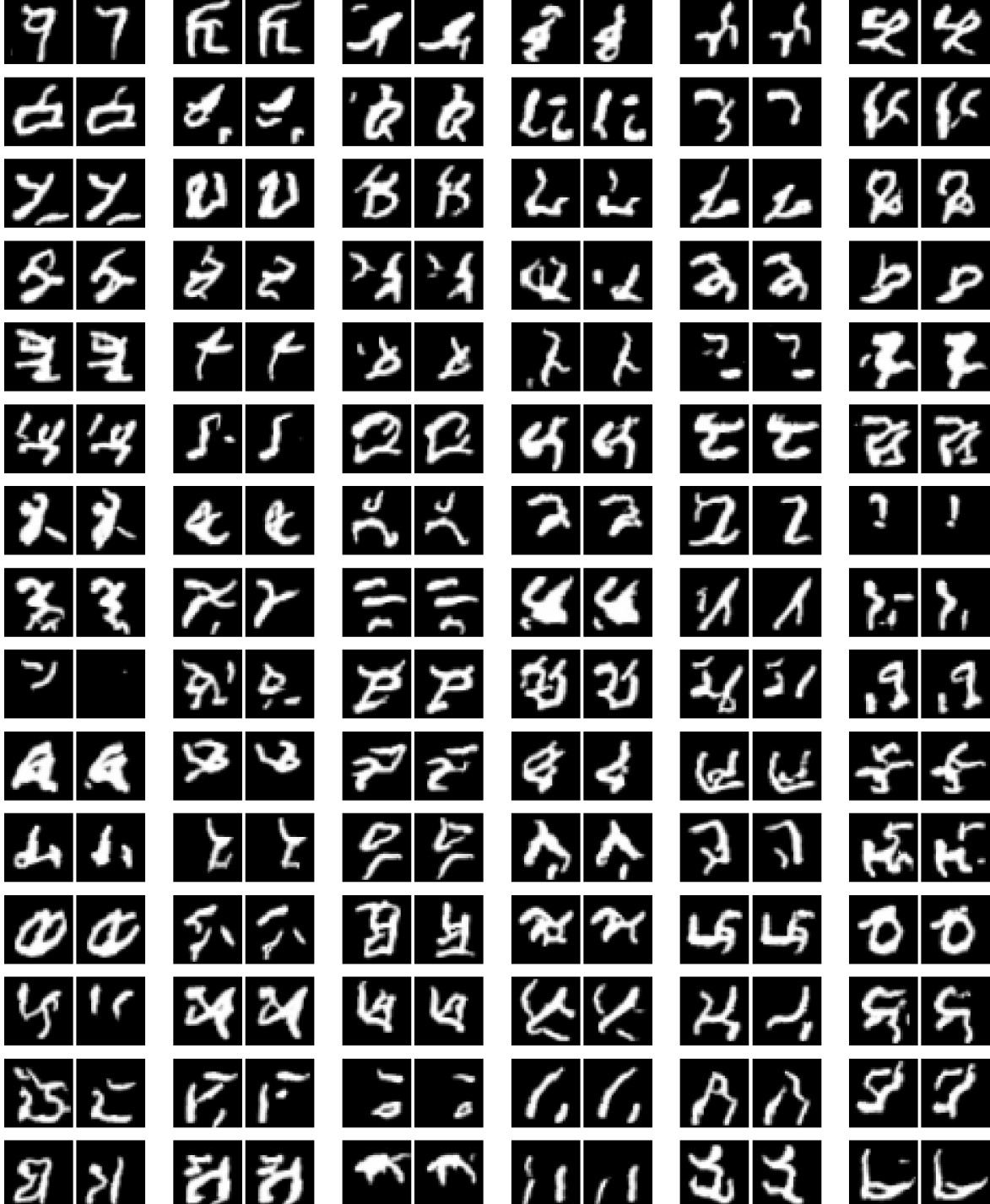

*Figure 13.* Further comparison between ResNet (right columns) and ELS Machine (left columns) samples for MNIST. Model is unconditional and has zero padding.

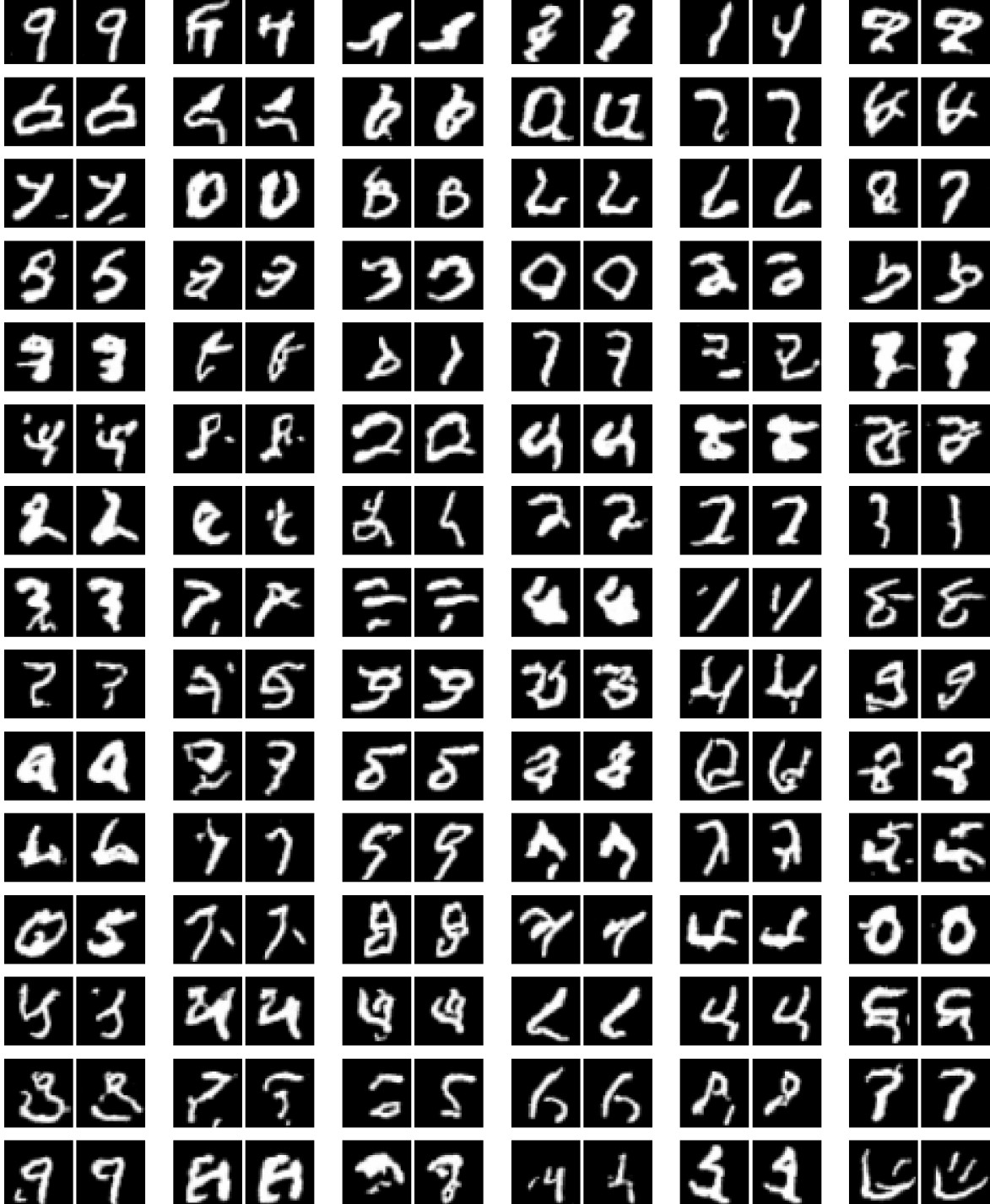

*Figure 14.* Further comparison between UNet (right columns) and ELS machine (left columns) samples for MNIST. Model is unconditional and has zero padding.

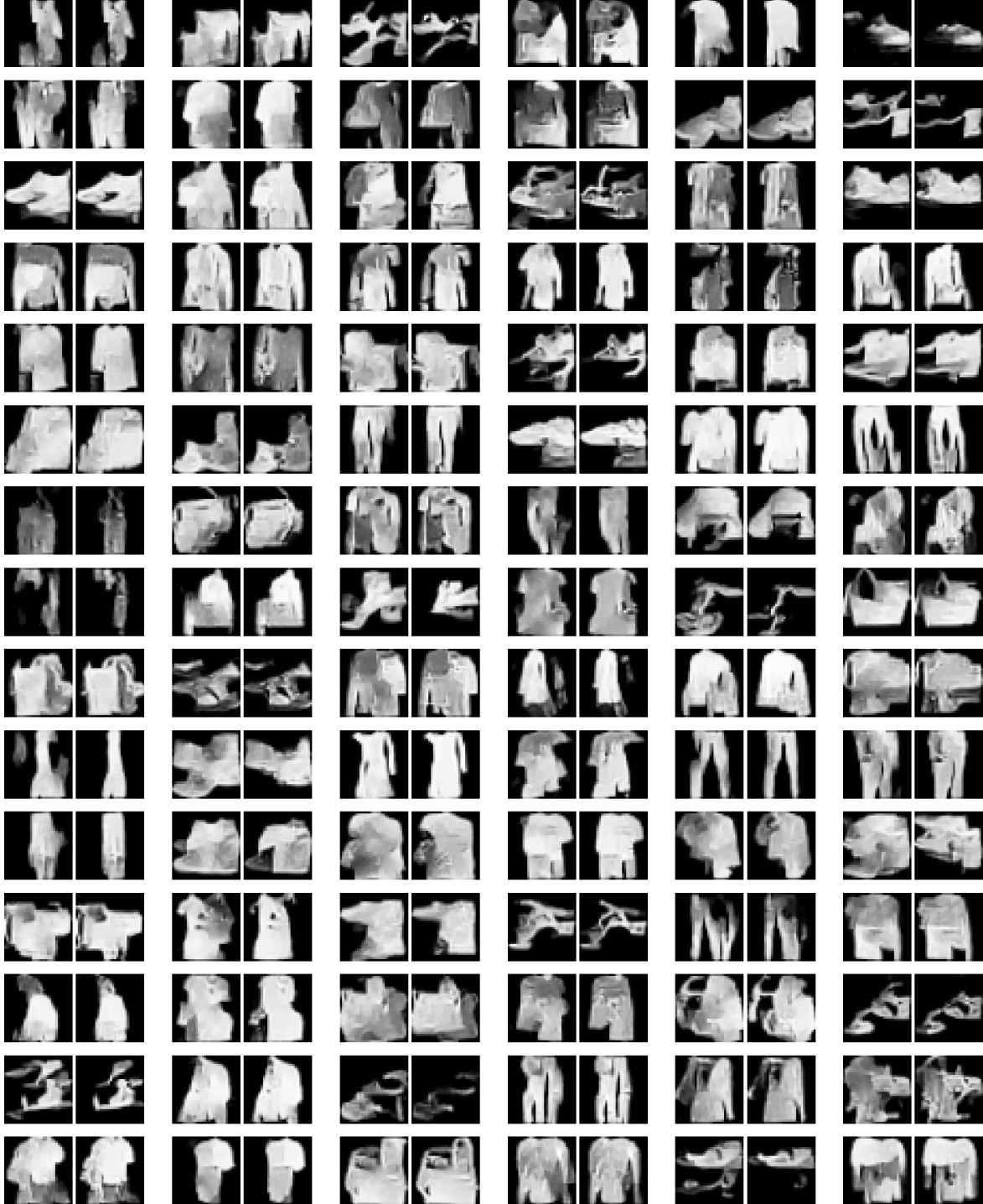

*Figure 15.* Further comparison between ResNet (right columns) and ELS Machine (left columns) samples for FashionMNIST. Model is class conditional and has zero padding.

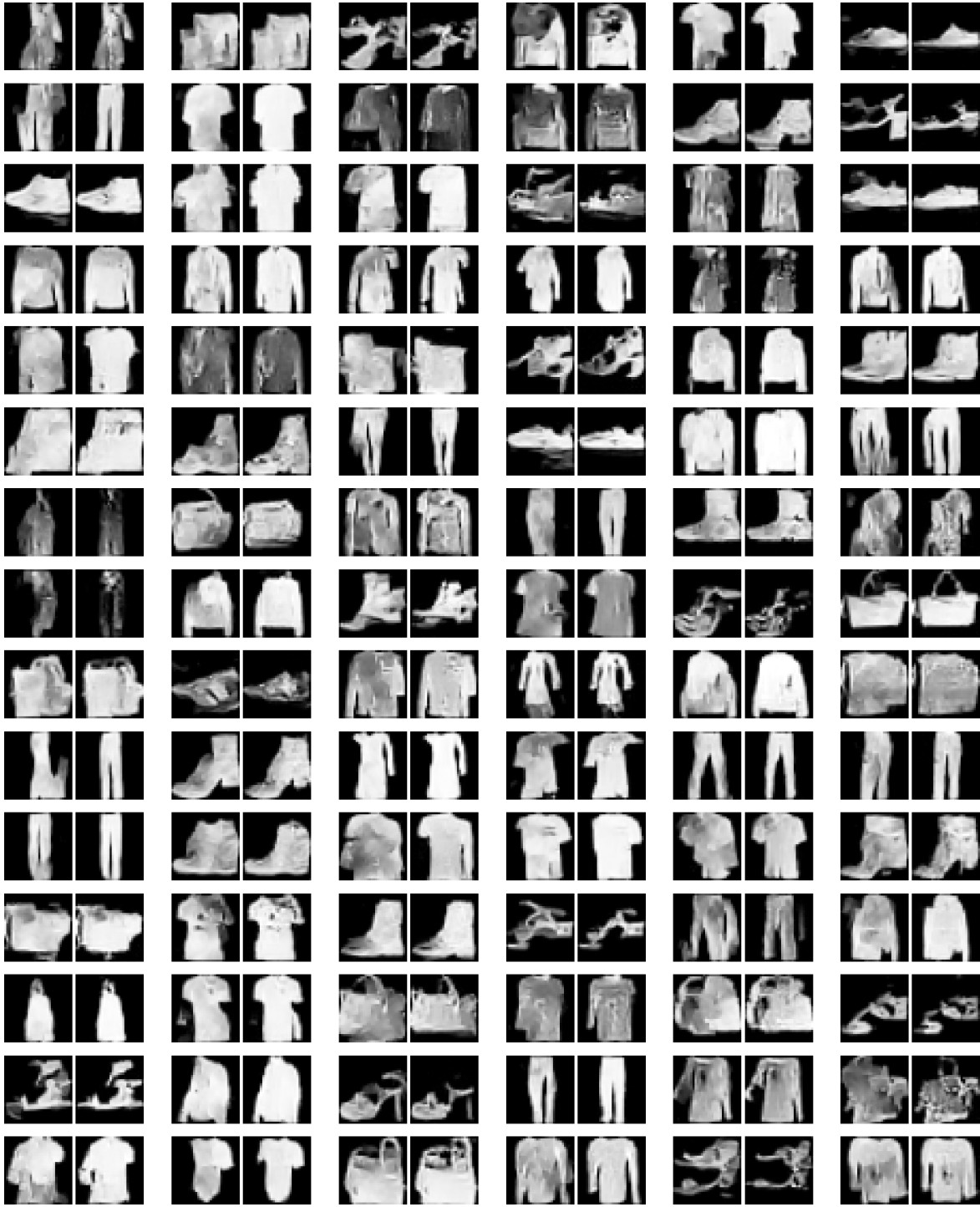

*Figure 16.* Further comparison between UNet (right columns) and ELS machine (left columns) samples for FashionMNIST. Model is class conditional and has zero padding.

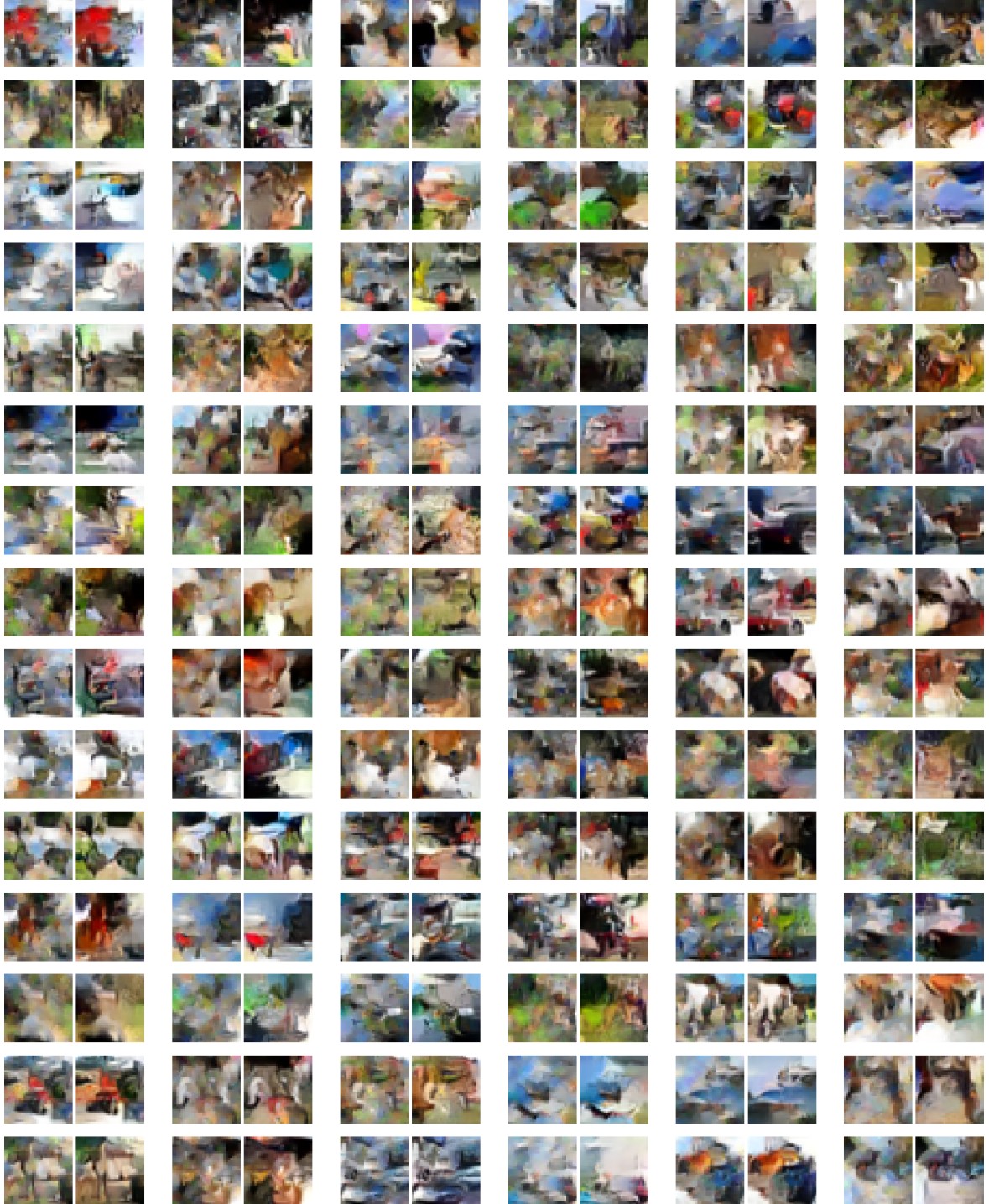

*Figure 17.* Further comparison between ResNet (right columns) and ELS machine (left columns) samples for CIFAR10. Model is class conditional and uses zero padding.

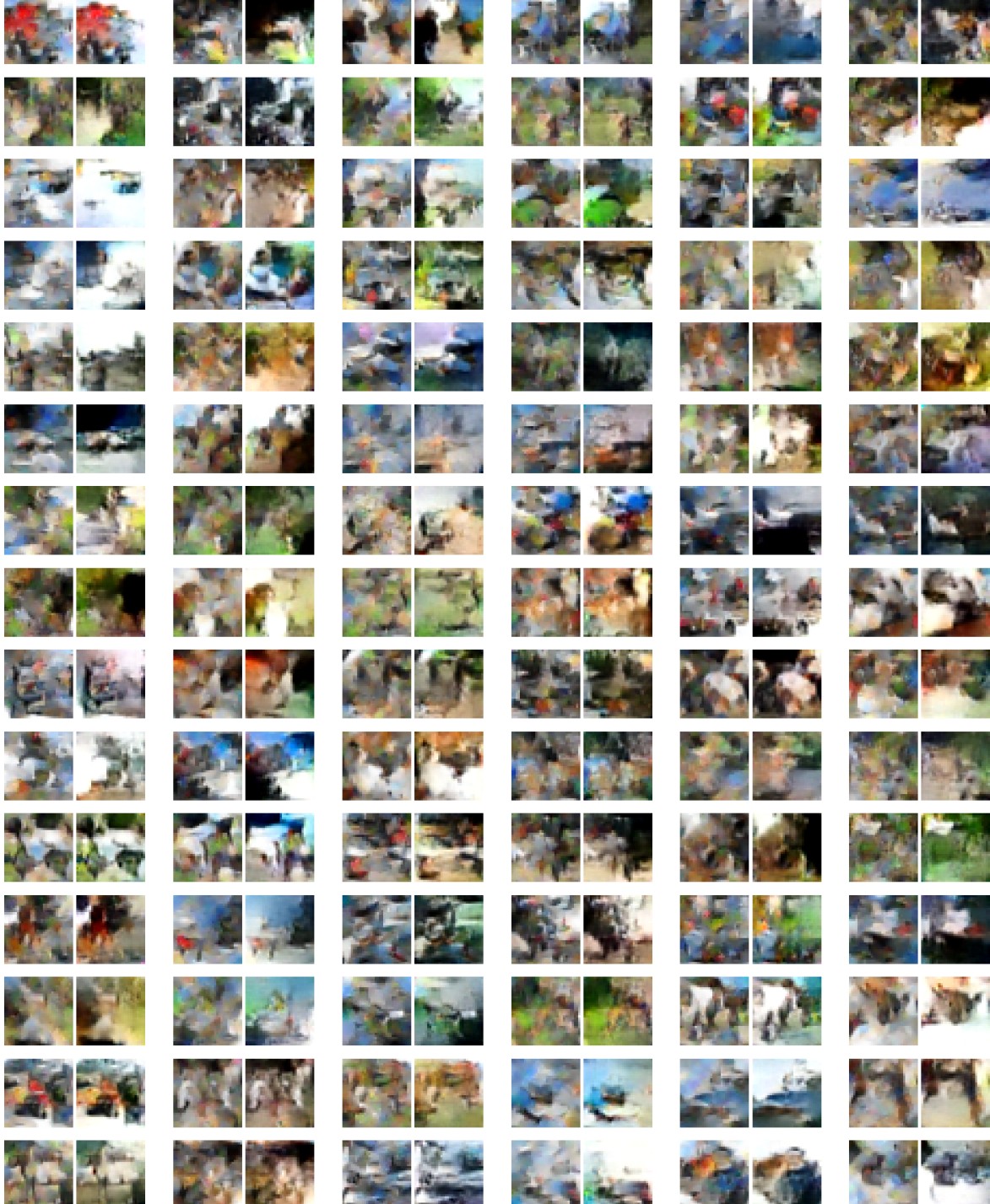

*Figure 18.* Further comparison between UNet (right columns) and ELS machine (left columns). Model is class conditional and has zero padding.

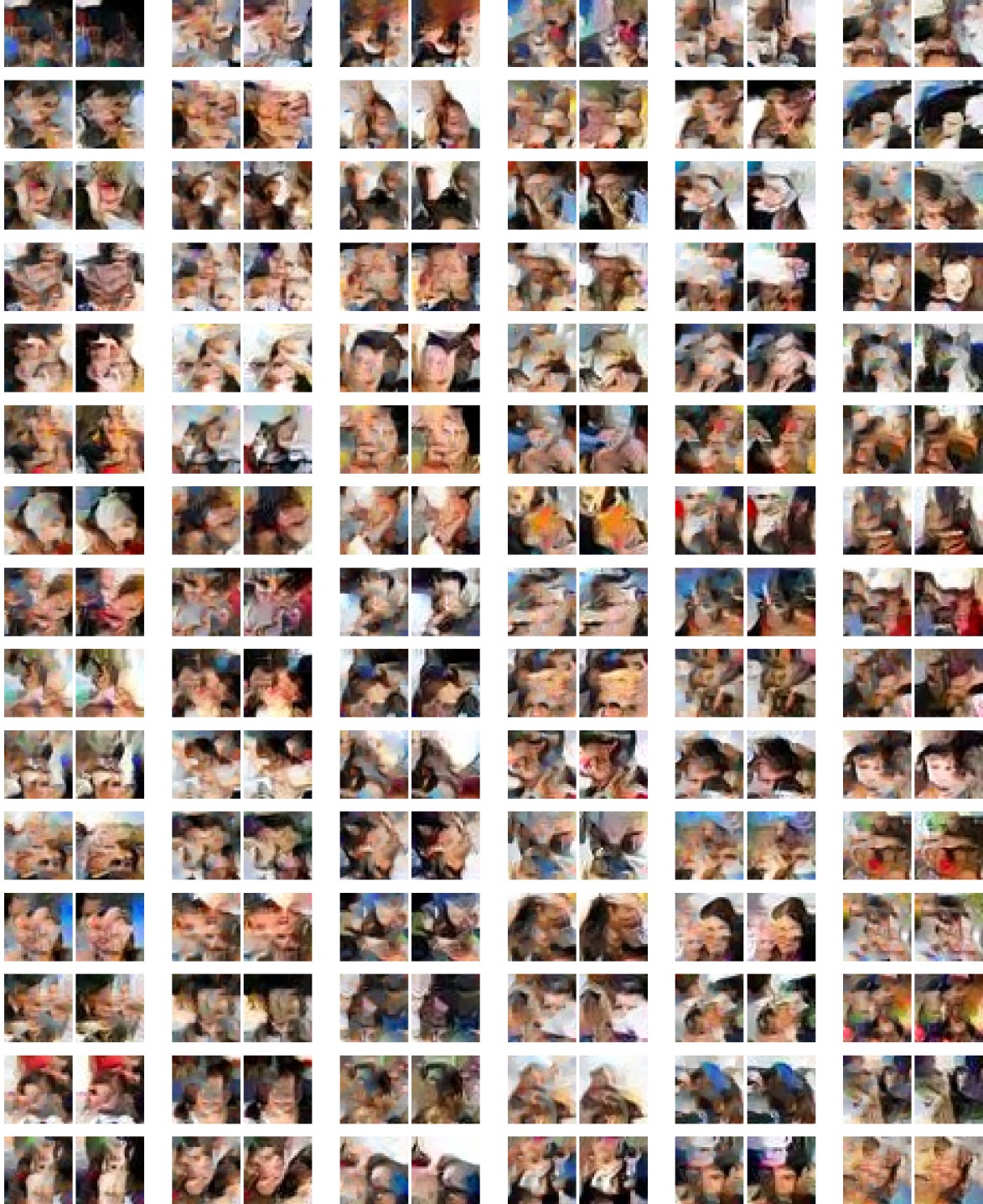

*Figure 19.* Further comparison between UNet (right columns) and ELS machine (left columns) on CelebA32x32. Model is class conditional and has zero padding. Model samples use 150 timesteps, ELS samples use 20.

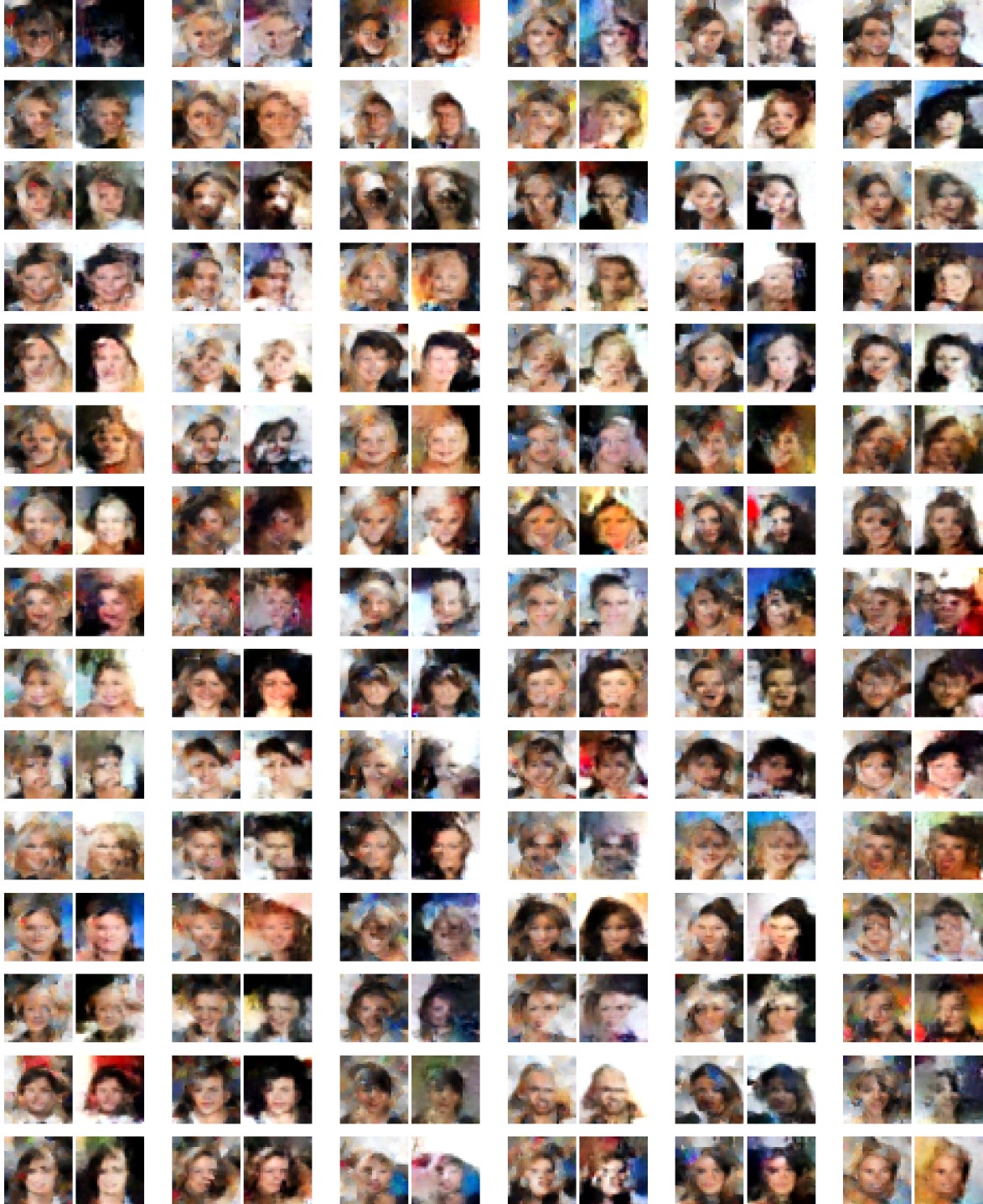

*Figure 20.* Further comparison between UNet (right columns) and ELS machine (left columns) on CelebA32x32. Model is class conditional and has zero padding. Model samples use 150 timesteps, ELS samples use 20.

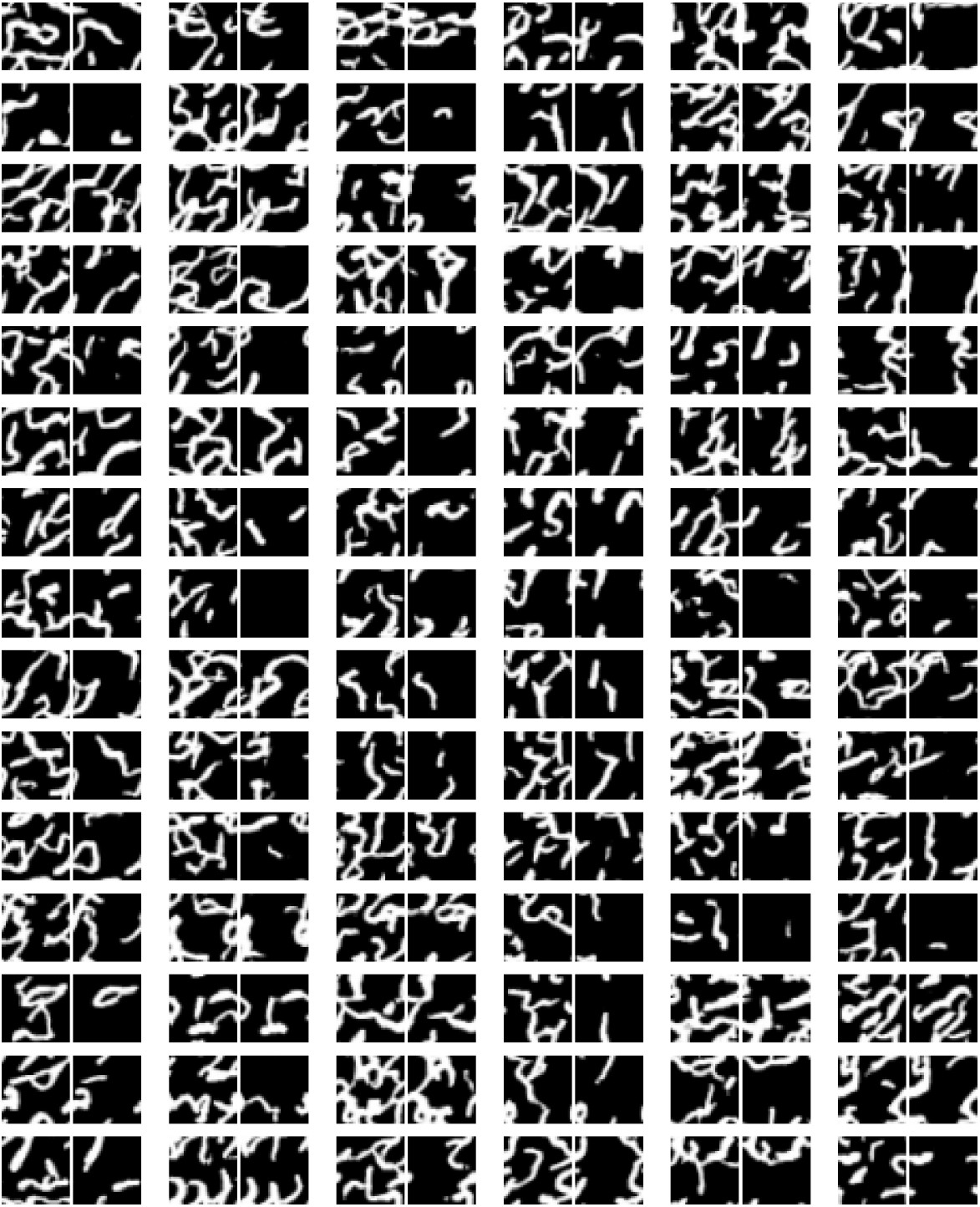

*Figure 21.* Further comparison between ResNet (right columns) and ELS machine (left columns) on MNIST. Model is unconditional and has circular padding.

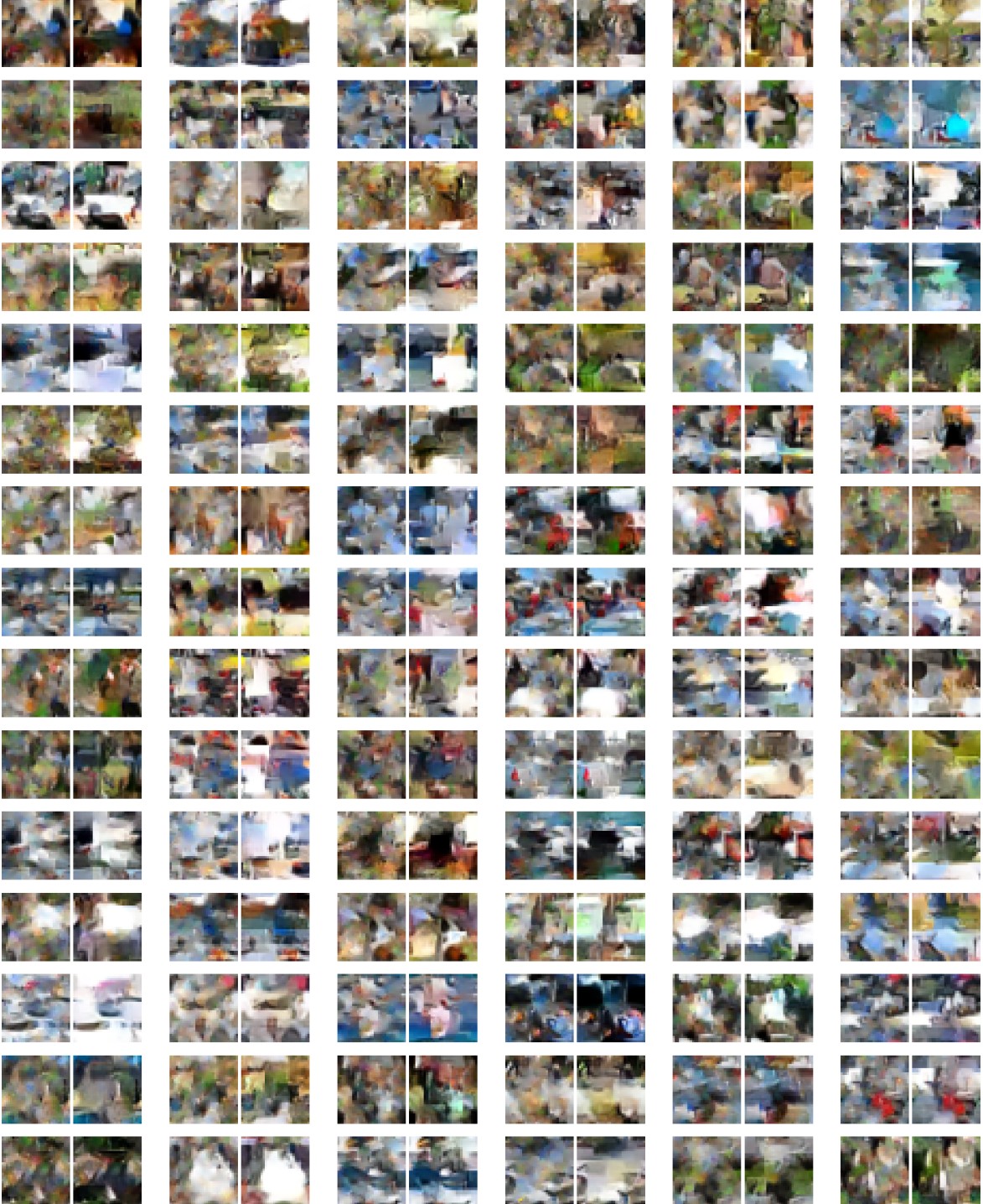

*Figure 22.* Further comparison between ResNet (right columns) and ELS machine (left columns) samples on CIFAR10. Model is class conditional and has circular padding.

(a) Incoherent output
(b) Non-matching output

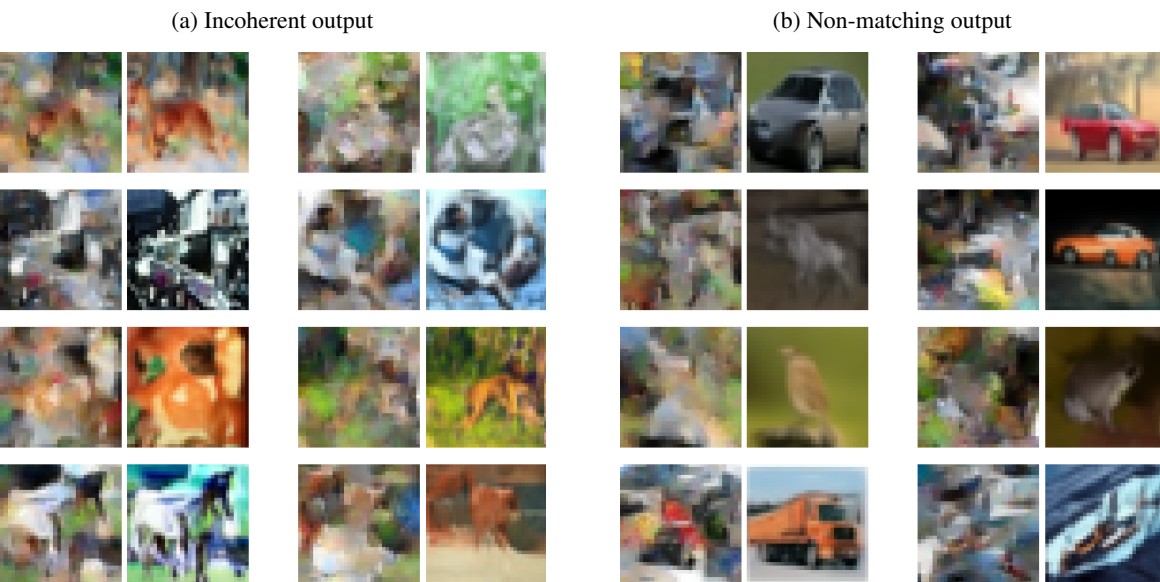

*Figure 23.* ELS Machine (left) vs. attention-enabled UNet (right) pairs. Panel (a) shows outputs where the Attentive UNet produces semantically incoherent, uninterpretable outputs, which tend to match strongly with the ELS Machine outputs. Panel (b) shows examples where the Attentive UNet produces samples not obviously matched with the ELS machine.

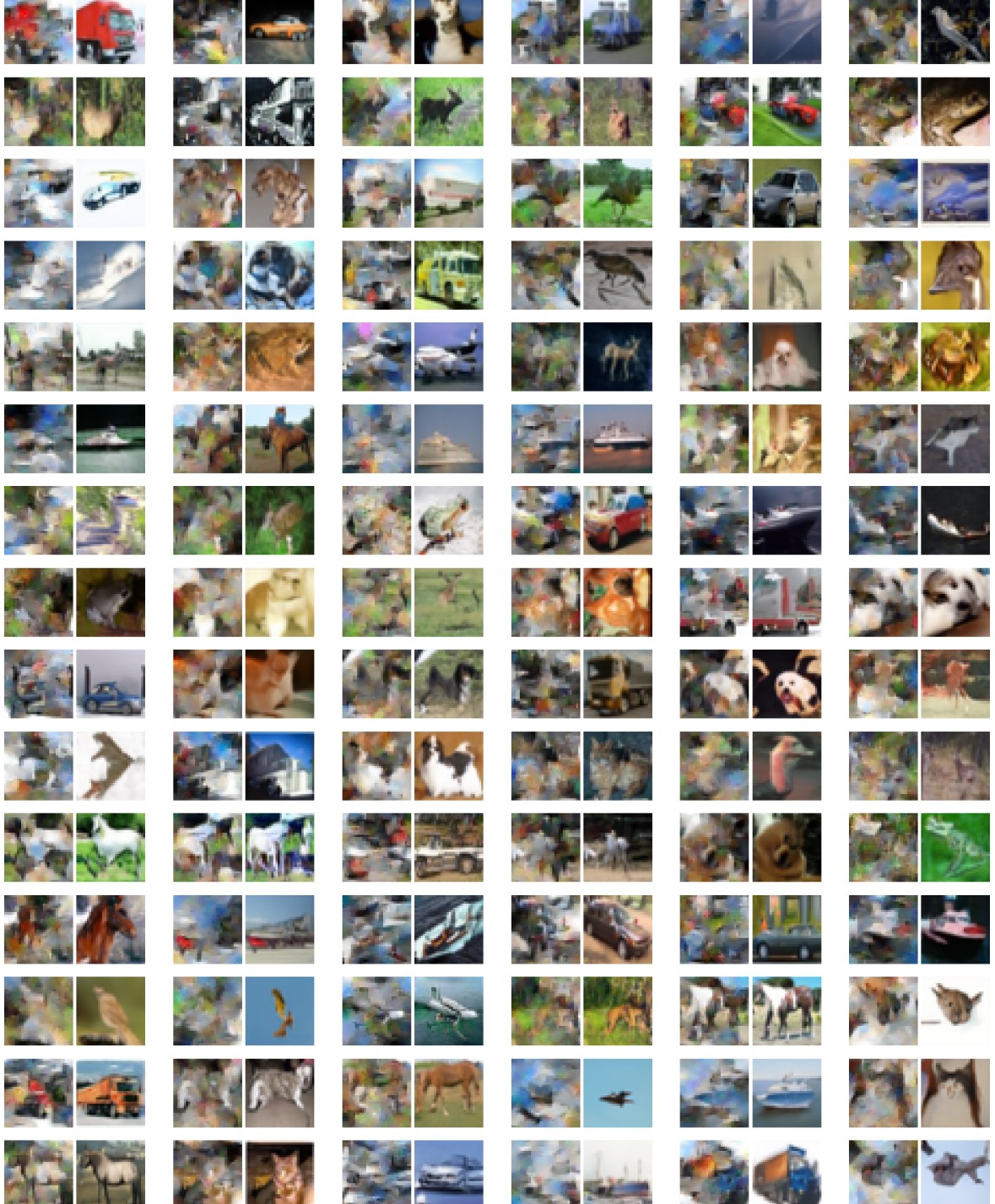

*Figure 24.* Further comparison between attention-enabled UNet (right columns) and ELS machine (left columns) samples on CIFAR10. Model is class conditional and has zero padding.

