# OpenReview forum: "An analytic theory of creativity in convolutional diffusion models"
_ICML.cc/2025/Conference — ICML 2025 oral_

### Official Review · Reviewer_qd8x · 2025-03-01

**Overall Recommendation:** 5

**Summary:**

The paper proposes a formula to predict images generated by convolutional diffusion models. The analysis suggests that biases in convolutional neural networks—such as locality and translational equivariance—prevent diffusion models from learning a perfect score function, encouraging them to generate samples that were not present in the training data.

**Claims And Evidence:**

The evidence is promising in the cases of MNIST and FashionMNIST. However, the evaluation based on CIFAR-10 is less conclusive. This is still acceptable because the ResNet-based diffusion model itself still struggles to generate high-quality images.

**Essential References Not Discussed:**

-

**Experimental Designs Or Analyses:**

Experimental design is straight-forward so there is no issue with it.

**Methods And Evaluation Criteria:**

r^2 measures how well the predicted outputs (ELS machine) correlate with the actual outputs (diffusion model).
This is pixel-level measurement. In the image generation, there are multiple choices of metrics that can be used along with pixel-level metrics, such as FID and Inception Score. The only concern is this work does not include these metrics.

**Other Comments Or Suggestions:**

-

**Other Strengths And Weaknesses:**

I cannot think of the weakness.

**Questions For Authors:**

No question

**Relation To Broader Scientific Literature:**

There are public debates about whether diffusion models copy art from humans.
This work theoretically demonstrates that diffusion models can be creative and generate something no human has seen before. This work is important in relation to the broader scientific literature, as it has the potential to extend the framework to other generative AI models, such as large language models (LLMs).

**Theoretical Claims:**

Although I did not go into the details of proofs, the high-level idea and motivation sounds promising.

---

> ### Author Rebuttal · Authors · 2025-04-01
>
> We thank the reviewer for their comments and suggestions. Below, we hope to carefully address a few of their concerns:
>
> >The evidence is promising in the cases of MNIST and FashionMNIST. However, the evaluation based on CIFAR-10 is less conclusive. This is still acceptable because the ResNet-based diffusion model itself still struggles to generate high-quality images.
>
> We would like to emphasize that the primary objective of our paper is to find a predictive theory for small convolutional neural networks, rather than a predictive theory of high performance networks which require more elements like transformers. While it is true that these networks are limited in their performance, particularly on CIFAR-10 (as the reviewer notes), this is a feature of the models we are attempting to study, and a theoretical model that obtained higher performance would not be an accurate description of the model class. Predicting the defects of models is in fact part of the objective of our paper-- for instance, we are interested in understanding the origin of common spatial consistency problems in diffusion models, as shown e.g. in figure 4.
>
> >r^2 measures how well the predicted outputs (ELS machine) correlate with the actual outputs (diffusion model). This is pixel-level measurement. In the image generation, there are multiple choices of metrics that can be used along with pixel-level metrics, such as FID and Inception Score. The only concern is this work does not include these metrics.
>
> While FID and Inception Score are commonly used metrics, they are measures of distributional similarity between large sets of unpaired images. However, the primary task in our paper is to measure the similarity between individual pairs of images, and thus FID and Inception Score are not appropriate for the task.  Indeed any *distributional* distance metrics is too weak for our purpose: we don’t just want to show that the ELS machine and the trained UNet/Resnet sample from similar distributions; we want to show a much stronger result that for every single noise realization, the particular image generated by the ELS machine, and the corresponding paired image generated by the UNet/ResNet is very similar.
>
> Feature-wise distances might be considered, but there is less of a standard consensus on the features to use for small black and white datasets such as MNIST and FashionMNIST (InceptionV3, from which FID is derived, was trained on ImageNet, whose statistics do not resemble the statistics of MNIST and thus whose features are unlikely to be useful for capturing the features of that dataset). Rather than look for a nonstandard feature space, we felt it was better to stick with a metric in the underlying pixel space, where the resulting numbers would be clearly interpretable.
>
> There are other pixelwise distance metrics that are standard in the literature that could be considered, such as L2 distance and cosine similarity. The former is not invariant to rescaling, and since we wanted to have a comparison between the performance of the ELS machine on models trained on image datasets with different statistics, we felt it was better to choose a scale-invariant metric such as r^2 or cosine similarity. These two metrics are very similar quantitatively, and we felt it would be redundant to include both. The choice of r^2 does not privilege our model; the values of the median cosine similarities for various model configurations are presented below for comparison:
>
> | Model                       | ELS/CNN Cosine Similarity |
> |-----------------------------|---------------------------|
> | MNIST/UNet/Zeros            | 0.93                      |
> | CIFAR10/UNet/Zeros          | 0.82                      |
> | FashionMNIST/UNet/Zeros     | 0.88                      |
> | MNIST/ResNet/Zeros          | 0.97                      |
> | MNIST/ResNet/Circular       | 0.82                      |
> | CIFAR10/ResNet/Zeros        | 0.89                      |
> | CIFAR10/ResNet/Circular     | 0.90                      |
> | FashionMNIST/ResNet/Zeros   | 0.92                      |
> | CIFAR10/UNet+SA             | 0.84                      |

---

> > ### Comment · Reviewer_qd8x · 2025-04-02
> >
> > Thank you for the clarification. I think this paper should be accepted without any concerns. I increase the score to be strong accept.

---

### Official Review · Reviewer_CGco · 2025-03-03

**Overall Recommendation:** 4

**Summary:**

This work proposes that biases of translation-equivariance and locality are sufficient to explain novel image generation in fully convolutional diffusion generative models. It does so by showing that a closed-form score model subject to those constraints qualitatively recapitulates the images generated by trained CNNs.

## Update after rebuttal

The authors' rebuttal answers my questions and concerns, and I remain strongly in favor of acceptance.

**Claims And Evidence:**

The claims are largely supported by convincing evidence. Given the broad interest in diffusion generative models, it goes without saying that this paper should be of substantial interest to the ICML audience, and may well be influential.

I would encourage the authors to reconsider their use of the term "creative", as I think "novel" or "original" would be somewhat more precise, and would be less burdened with humanistic baggage. For instance, the paper could alternatively be titled "An analytic theory of novel image generation in convolutional diffusion models". However, this is at least partially a matter of taste, and I leave the decision to the authors' discretion.

**Essential References Not Discussed:**

I think the authors do a good job addressing relevant literature on theories for how diffusion models generate novel images; no omissions stood out.

**Experimental Designs Or Analyses:**

The experiments are largely well-designed. One concern is that all quantitative comparisons use pixelwise $r^2$, and no clear justification is provided for this choice.

**Methods And Evaluation Criteria:**

The methodology seems sound.

**Other Comments Or Suggestions:**

- There are a number of in-text citations for which \citet should be used in place of \citep, e.g. when citing Kadkhodaie et al. in Line 98.

- Please use a 1:1 aspect ratio in Figures 8-10.

- It would be helpful if the authors swapped the column ordering of Figure 19 so that the ELS machine is on the right like in other figures.

**Other Strengths And Weaknesses:**

I think the authors should state more clearly in the Introduction that the excellent predictions resulting from their closed-form model require calibration of the time-dependent patch scale based on the quality of the predictions. I don't think this is a substantial limitation - after all this is only a few parameters - but I do think it's important to mention clearly, if only to highlight that the factors determining the schedule of scale decreases is an interesting topic for future work.

**Questions For Authors:**

- The finding that imposing circular boundary conditions leads to more texture-like generated images is interesting. Can you provide any intuition for why that is? Does it arise purely through interactions between the artificially-joined boundary patches?

- What happened in the ELS-generated image in the eigth row of the second column of Figure 17? Here the model has produced an image very different from the convolutional model, and indeed one that lacks the expected localized features.

- This is something that could be left to future work, but I am curious if the ELS model could recapitulate the results of the forward-backward experiments performed by Sclocchi, Favero, & Wyart. In particular, could some of the transition points they detect be linked to changes in the patch scale?

**Relation To Broader Scientific Literature:**

This paper is related to the broader literature through the position of the literature on diffusion models within the broader machine learning field.

**Theoretical Claims:**

Most of the derivations in this paper are quite straightforward. The main theoretical result is Theorem 4.1; its proof appears correct.

---

> ### Author Rebuttal · Authors · 2025-04-01
>
> We thank the reviewer for their highly detailed feedback and insightful suggestions. Below, we carefully address several of the points that they raised.
>
> >I would encourage the authors to reconsider their use of the term "creative", as I think "novel" or "original" would be somewhat more precise, and would be less burdened with humanistic baggage […] However, this is at least partially a matter of taste, and I leave the decision to the authors' discretion.
>
> We appreciate the reviewer’s concern about this choice of nomenclature. While a matter of taste, we did arrive at the term by trying to encapsulate a particular technical phenomenon; roughly, “originality subject to constraints embedded in the data.” We felt that it was important to distinguish this behavior from the trivial originality type of e.g. a white noise sampler. In order to make this case clearer, we will add language in our revisions that makes more explicit the particular characteristics that we are attempting to capture with this word choice.
>
> The term that we felt was otherwise nearest to the concept we were trying to describe is “generalization.” While a less humanistic word choice, we felt that this term too had baggage that we were not totally comfortable with. Generalization is often used to mean something along the lines of “correct generalization” or the gap between training and test set performance. It was not clear to us that this was the right notion for the phenomena in our paper (e.g. the outputs of Figure 5 are "incorrect"); as such, we elected to sidestep this by instead using a nontechnical term.
>
> >One concern is that all quantitative comparisons use pixelwise r2, and no clear justification is provided for this choice.
>
> The reasoning behind the choice of the r2 metric is as follows. The task we are trying to benchmark is a paired image comparison task, and thus taking the median of some image distance metric across the sample set seemed appropriate. Many metrics commonly used in the generative image model literature, such as FID, are *distributional* distance metrics and so we did not consider them as they were too weak for our purpose. Distances between pairs of images in the underlying pixel space seemed both most natural to us and also the most stringent possible test for our model. We considered using the L2 distance directly, however, this metric is sensitive to the overall scale of the image distribution; since we wanted a metric that would allow us to compare the performance across different datasets that have different image statistics, the metric should be invariant to overall scale. The r2 metric is a standard metric which satisfies these criteria. We also felt that the 0-1 scores that the r2 metric assigns were more intuitively understandable than the raw l2 scores.
>
> Another alternative metric choice that satisfied the criteria outlined above was cosine similarity. This metric is very similar to the r2 metric, and the results are given below:
>
> |Model|ELS/CNN Cos|
> |---|---|
> |MNIST/UNet|0.93|
> |CIFAR10/UNet|0.82|
> |FMNIST/UNet|0.88|
> |MNIST/ResNet/Zeros|0.97|
> |MNIST/ResNet/Circular|0.82|
> |CIFAR10/ResNet/Zeros|0.89|
> |CIFAR10/ResNet/Circular|0.90|
> |FMNIST/ResNet/Zeros|0.92|
> |CIFAR10/UNet+SA|0.84|
>
> Ultimately, rather than presenting two separate but highly related metrics, we made the choice to use the r2 metric alone.
>
> >The finding that imposing circular boundary conditions leads to more texture-like generated images is interesting. Can you provide any intuition for why that is? Does it arise purely through interactions between the artificially-joined boundary patches?
>
> The intuition for the texture-like behavior is that, without any global positional information, local models are not able to robustly coordinate their generation in order to produce coherent images. Roughly, each independent part of the denoised image spontaneously decides to resemble a totally random location in a training image; this naturally produces a bit of a jumble. Adding borders helps coordinate this generation process by ‘pinning down’ the boundary. This helps especially for datasets such as MNIST and FashionMNIST, where the boundary is very regular (a stereotyped black background).
>
> >What happened in the ELS-generated image in the eigth row of the second column of Figure 17?
>
> This was a plotting bounds issue; the image should appear visually monochromatically black and has been fixed.
>
> >This is something that could be left to future work, but I am curious if the ELS model could recapitulate the results of the forward-backward experiments performed by Sclocchi, Favero, & Wyart.
>
> We thank the reviewer for this interesting suggestion. We suspect that there may be a connection, and in particular we are curious in particular about whether the ELS machine framework might shed light on what the “natural” analogue of the hierarchy of variables that they study might be in real datasets. However, we will leave detailed investigation up to future work.

---

> > ### Comment · Reviewer_CGco · 2025-04-01
> >
> > Thank you for the response, which addresses my concerns. I will maintain my score, as this paper should certainly be accepted.

---

### Official Review · Reviewer_TYVX · 2025-03-07

**Overall Recommendation:** 5

**Summary:**

This paper develops a theory for why convolutional diffusion models fail to learn the ideal score function. It is theorized that this is due to locality from small receptive fields and translational equivariance. Under these assumptions, an optimal minimum MSE approximation to the ideal score function is derived subject to locality and broken translational equivariance constraints. The so-called equivariant local score machine shows that convolutional diffusion models compose different patches from training examples together, which the papers calls a locally consistent patch mosaic. There is very high quantitative (r^2 values) agreement between outputs from convolutional diffusion models and the ELS machine, suggesting that the ELS provides a plausible explanation for the mechanisms of convolutional diffusion models. Some of the theory also holds up when self-attention is introduced.

## Update after rebuttal
I maintain my score of strong accept. I thank the authors for their effort in addressing my comments, especially the Celeb-A experiments.

**Claims And Evidence:**

The claims are strongly supported by theory and empirical results, showing strong agreement between outputs from the ELS machine and from convolutional diffusion models.

**Essential References Not Discussed:**

The paper has done due diligence to cite related works such as Kadkhodaie et al., 2023a and concurrent work (Niedoba et al., 2024). So, I do not see any issues in missing essential references.

**Experimental Designs Or Analyses:**

The experimental designs and analysis are sound. The analysis of spatial inconsistencies from excess late-time locality is especially insightful, providing an explanation for the well-known phenomenon that diffusion models struggle in generating limbs.

**Methods And Evaluation Criteria:**

The proposed methods and evaluation criteria make sense. r^2 is measured between the outputs from the ELS machine and convolutional diffusion models to measure agreement.

**Other Comments Or Suggestions:**

In the Supplementary Section D. Samples, the page is blank followed by samples in the next page.

**Other Strengths And Weaknesses:**

**Strengths**
* I think this is a very strong paper that can have big impact since it provides a theory for why convolutional diffusion models can generalize. This has implications for data attribution, fixing artifacts from diffusion model samples, improving adherence to conditioning, etc. The paper is also very well-written. As a non-theorist, the development of the ELS machine was very intuitive, and I appreciated the illustrations such as Figure 3 to build and intuition. I provide some small suggestions below.

**Suggestions**
* How important is each constraint in the analytic solution? It is mentioned in the paper that the equivariance constrained machine can only "generate training set images globally translated to any other location." Is it possible to ablate each constraint and observe the correlation with the diffusion samples?
* I think Fig. 1 can be further improved by including a column of the input noise. This will emphasize the message that the analytic theory predicts almost the same output as a convolutional diffusion model given the same input noise.
* I understand that the theory can only mainly explain toy, small-scale settings such as MNIST, CIFAR, Fashion-MNIST. But there is not much compositionality in these datasets. CIFAR is a difficult dataset to generate cohesive images, so it is hard to see any "creativity" since it is difficult to see any semantics in the images. If possible, I think applying the ELS machine on CelebA-64x64 (or even 32x32 and grayscale) can produce very convincing results. It is a fairly homogeneous dataset, so fitting a convolutional diffusion model should not be difficult. Also, there is a lot of room for compositionality since facial expressions (e.g., smile) or accessories (e.g., glasses) can be patches that are combined with other patches to demonstrate "creativity." I think this would make it more convincing beyond the theory community.

**Questions For Authors:**

Please see **Suggestions** above.

**Relation To Broader Scientific Literature:**

The paper gives care to discussing previous work that shows that the ideal diffusion model should memorize its data. This provides a nice setup for the contributions of this paper, which explain why convolutional diffusion models don't in fact memorize their data, and essentially compose patches from different training examples.

**Theoretical Claims:**

The theoretical claims seem to be correct. Ultimately, the empirical results show that the derived analytic solution closely matches with the true outputs from convolutional diffusion models.

---

> ### Author Rebuttal · Authors · 2025-04-01
>
> We thank the reviewer for their detailed feedback and suggestions. We address some of these below:
>
> >How important is each constraint in the analytic solution? […] Is it possible to ablate each constraint and observe the correlation with the diffusion samples?
>
> An equivariant score machine on its own will only generate memorized training examples, similar to the ideal score machine, but translated a random amount. This is highly discrepant from the observed data, especially in the presence of boundary conditions that fix the spatial position of at least the edges of the image. Due to time constraints we have not yet performed the ablation of locality; however, we expect a priori for the reason outlined before that it will perform uniformly lower across all categories than even the ideal-score baseline.
>
> To address the part of the reviewer’s question about the performance of the locality constraint without equivariance, we discuss several aspects. Firstly, fully-equivariant circularly-padded models display features clearly inconsistent with patches knowing their position. This is shown clearly in the samples from the circularly-padded ResNet on MNIST, which include line segments that intersect the boundary, a feature that is not present in any training example in MNIST and cannot be elicited from the LS Machine.
>
> Secondly, we evaluated the correlation between the outputs of each model we studied in the paper and the outputs of a corresponding LS Machine. The quantitative results of these results are summarized in the table below:
>
> |Model|LS/CNN r^2|ELS/CNN r^2|
> |---|---|---|
> |MNIST/UNet/Zeros|0.83|0.84|
> |CIFAR10/UNet/Zeros|0.80|0.82|
> |FashionMNIST/UNet/Zeros|0.91|0.91|
> |MNIST/ResNet/Zeros|0.84|0.94|
> |MNIST/ResNet/Circular|0.33|0.77|
> |CIFAR10/ResNet/Zeros|0.86|0.90|
> |CIFAR10/ResNet/Circular|0.80|0.90|
> |FashionMNIST/ResNet/Zeros|0.88|0.90|
> |CIFAR10/UNet+SA|0.74|0.75|
>
> We find that the ELS machine uniformly outperforms the LS machine for ResNets, and performs above or at par with the LS machine for UNets, but the discrepancy is small for zero-padded models, indicating that locality is the main factor. Qualitatively, however, the LS machine samples are "grainy" and the ELS samples are visually much better, which is not reflected in this metric.
>
> We found one outcome, in response to the suggestion that we study CelebA, which did not follow the trend described above. We describe the results below.
>
> >If possible, I think applying the ELS machine on CelebA-64x64 (or even 32x32 and grayscale) can produce very convincing results.
>
> We strongly thank the reviewer for this suggestion. To address it, we trained a ResNet and a UNet model on CelebA-32x32 grayscale. Our analysis is not yet complete due to the large computational expense required for the calibration and generation process, but we have been able to perform a preliminary analysis of the LS Machine and ELS Machine, using a) a reduced dataset and b) manually calibrated scales. In these experiments, we have so far found the following.
>
> 1) The ELS machine matches the ResNet model, with median r^2 ~ 0.96. However, qualitatively, the samples produced by the ResNet are of poor quality, similarly to CIFAR10.
> 2) The UNet model trained on this dataset is able to produce recognizable faces. We found, to our surprise, that it appears to be better fit by the *LS Machine* (r^2 ~ 0.92) as opposed to the ELS Machine (r^2 ~ 0.90). The LS Machine captures the placement of noise, eyes, and mouth in the image; these details are not captured by the ELS machine, which generates similar images overall but which lacks these key human-interpretable features.
>
> We believe that the explanation for this behavior is as follows. The ResNet is shallow and has a *hard* locality constraint. The UNet receptive field size is formally very large and includes the image border everywhere; the fact that it exhibits local and equivariant behavior is an *emergent phenomenon* that requires pixels to use less information than maximally possible. Restoring positionality while preserving locality is akin to reintroducing some but not all of the information that is discarded.
> More experiments will be needed to study this capability, but we believe that this will not be fully achievable within the timeframe of the revisions and defer this to future work. However, we intend to incorporate the additional findings on CelebA into our paper in the final revision.
>
> >I think Fig. 1 can be further improved by including a column of the input noise.
>
> We could in principle pick images for the figure such that the same initial noises could be used for black and white images; however, CIFAR10 images have 3 channels, while FMNIST and MNIST have only one, so we would always need at least two additional columns of noise. We feel that the addition of these multiple columns would dilute the visual impact of the initial figure; thus, if the reviewer agrees, we would like to retain the figure design for figure 1.

---

> > ### Comment · Reviewer_TYVX · 2025-04-03
> >
> > I thank the authors for their response. The CelebA results will be a nice addition to the paper. While more experiments to validate your hypothesis for why the LS Machine outperforms the ELS Machine for the UNet are not necessary, the speculation on why this phenomenon may be occurring is still appreciated.

---

> > > ### Author Response · Authors · 2025-04-07
> > >
> > > We thank the reviewer for their comment.
> > >
> > > We would like to make two additional comments with regards to recent results that we found while performing an additional ablation experiment during the recent revisions process.
> > >
> > > Firstly, we were able to achieve high performance on CNNs trained on Celeba32x32 with color, and intend to report those results instead of the results on the grayscaled dataset. We are unfortunately unable to report the final correlation numbers for this dataset at this point pending completion of calibration, but preliminarily it looks like the performance of the ELS and LS models on ResNet/UNet respectively should be in line with the performance on the CIFAR10 models.
> > >
> > > Secondly, we had initially picked a relatively short set of timesteps (20) for our reverse process integration, and used the same timesteps for both the ELS machine and the compared CNNs. The primary justification for this decision was the large computational cost of running the ELS machine. We assumed that the highest fidelity theory/experiment agreement would be between the trajectories simulated with the same number of timesteps for both the ELS and the CNN model.
> > >
> > > We found however in a recent ablation that we could increase the median theory/experiment correlations across the board, including by up to 8%, using a much finer discretization (150 timesteps) for the CNNs, while keeping the ELS timesteps fixed at 20. We have not attempted to compute ELS machine outputs with a similarly large number of timesteps, which would require a large amount of computational resources.
> > >
> > > The specific quantitative values are as follows:
> > > |Model|ELS/CNN r^2 (150 steps)|ELS/CNN r^2 (20 steps)|
> > > |---|---|---|
> > > |MNIST/UNet/Zeros|0.89|0.84|
> > > |CIFAR10/UNet/Zeros|0.90|0.82|
> > > |FashionMNIST/UNet/Zeros|0.93|0.91|
> > > |MNIST/ResNet/Zeros|0.94|0.94|
> > > |MNIST/ResNet/Circular|0.77|0.77|
> > > |CIFAR10/ResNet/Zeros|0.95|0.90|
> > > |CIFAR10/ResNet/Circular|0.94|0.90|
> > > |FashionMNIST/ResNet/Zeros|0.94|0.90|
> > > |CIFAR10/UNet+SA|0.77|0.75|
> > >
> > > We intend to report both the older numbers as well as these newer "high-compute" numbers.

---

### Official Review · Reviewer_svst · 2025-03-21

**Overall Recommendation:** 5

**Summary:**

This paper presents an analytic theory of generalization in convolutional diffusion models. It identifies that, given a finite empirical dataset, the optimal score function produces a perfect reverse diffusion process, leading to replicas of training samples. The paper then hypothesizes that the creativity of real trained diffusion models arises from the inductive biases of parametric models (e.g., convolutional neural networks). Assuming equivalence and locality properties of the score function in image generation, the authors derive an analytic score model. This enables a case-by-case study of parametric score models (e.g., CNNs, ResNets), showing impressive correlation with actual trained models. This work provides a transparent interpretation of how diffusion models generate novel images.

**Claims And Evidence:**

The claims in this paper are clear and supported by theoretical derivations as well as convincing empirical evidence.

**Essential References Not Discussed:**

Several existing works share the intuition of memorization in ideal score functions:

- Gu, Xiangming, et al. "On memorization in diffusion models." arXiv preprint arXiv:2310.02664 (2023).
- Somepalli, Gowthami, et al. "Diffusion art or digital forgery? investigating data replication in diffusion models." Proceedings of the IEEE/CVF conference on computer vision and pattern recognition. 2023.

**Experimental Designs Or Analyses:**

The experimental design and analyses are sound. Perhaps one additional aspect could be explored: in real scenarios, two factors influence the final generation of diffusion models—the inductive biases inherent in the model family (e.g., U-Net) and those introduced by training dynamics (e.g., initialization, optimizers). I wonder whether there is a non-monotonic trend in $r\^2$ during the training process, which could provide insight into how optimization choices introduce implicit inductive biases.

**Methods And Evaluation Criteria:**

The evaluation primarily rely on $r\^2$, as the main goal of this paper is an analytic interpretation of generation in diffusion models. The $r\^2$ metric along with visual comparisons, show strong correlation between the theory-predicted results and actual generations from trained parametric diffusion models.

**Other Comments Or Suggestions:**

Typos:
- In Eq. (5): $\\sqrt{\\bar{\\alpha}\_t}\\varphi(x)-\phi$ should be $\\sqrt{\\bar{\\alpha}\_t}\\varphi(x)-\phi(x)$.

**Other Strengths And Weaknesses:**

This work is particularly interesting as it systematically investigates the failure of the ideal score function and studies the problem from a principled perspective. It also suggests the possibility of nonparametric generative models for high-dimensional data—provided we identify the correct inductive biases. The paper is well-written and convincing, with well-designed notations (although not consistent with common conventions).

One weakness is that while the theory predicts generation to some extent, there remains a clear quality gap. It is unclear whether an (approximate) theoretical solution is still feasible when considering more complex inductive biases, such as attention mechanisms.

**Questions For Authors:**

The method for determining patch sizes (e.g., Figure 4) appears somewhat heuristic. Did you attempt to derive it theoretically or formulate it as an optimization problem (e.g., minimizing the score-matching loss through cross-validation)? This could eliminate the need to rely on studying the saliency map of a pretrained model.

**Relation To Broader Scientific Literature:**

This work focuses on the generalization of generative models, which is a core topic in generative AI.

**Theoretical Claims:**

Yes, I checked most of the proofs (e.g., the derivation of the ELS machine), which appear correct.

---

> ### Author Rebuttal · Authors · 2025-04-01
>
> We appreciate the reviewer’s feedback and the thoughtful suggestions that they made for our paper. Below we address some of these comments and suggestions.
>
> >Perhaps one additional aspect could be explored: in real scenarios, two factors influence the final generation of diffusion models—the inductive biases inherent in the model family (e.g., U-Net) and those introduced by training dynamics (e.g., initialization, optimizers). I wonder whether there is a non-monotonic trend in r2 during the training process, which could provide insight into how optimization choices introduce implicit inductive biases.
>
> In response to the reviewer’s questions about the importance of training dynamics and initialization in determining model outputs, we retrained our models several times with identical data but different initial seeds, and evaluated the dynamics of the r2 metric with the ELS outputs.
>
> We found that the resulting generated outputs did not significantly vary between initialization seeds, with the median CNN-CNN pixelwise r2 between different post-training models achieving 0.9-0.97 on different datasets and architectures. We found that we could improve this somewhat by continuing training further and continuing to anneal the learning rate; e.g., training 600 epochs instead of 300 while continuing to decay the learning rate produced correlations typically > 0.95.
>
> We found that the ELS/CNN r2 was somewhat oscillatory over short time ranges (if learning rates remained too high near the end of training), but the overall trend was monotonically upwards across the training process. The oscillations we attributed to an instability in the overall image intensities of the model-generated output during the training process, which eventually disappeared as we annealed the learning rate appropriately. This dynamical effect did not seem to affect the output of the optimization process provided the learning rate was decayed sufficiently over the course of the training process. We intend to defer a more detailed study to further work.  But overall there is no strong indication that early stopping consistently helps with achieving a higher r^2 for these datasets.
>
> >The method for determining patch sizes (e.g., Figure 4) appears somewhat heuristic. Did you attempt to derive it theoretically or formulate it as an optimization problem (e.g., minimizing the score-matching loss through cross-validation)? This could eliminate the need to rely on studying the saliency map of a pretrained model.
>
> While we considered alternative options, we observed in our work (as shown in figure 4b) that the optimal patch size differed between the UNet model and the ResNet models, which showed that the optimal patch size for each model could not be an intrinsic statistical characteristic of the dataset and therefore could not be completely deduced through a process that did not use additional characteristics of the post-training models.
>
> We would like to emphasize that the approach that we took to calibrate our theory to the model did not actually use the saliency maps of the post-training models, which were shown in figure (4a) primarily as ancillary evidence for the observed multi-scale behavior. Rather, we performed a direct fit for each time by selecting the maximally performant ELS scale at each time step on a separate validation set.  Performance was defined as the correlation between the ELS machine’s predicted score function at each time, and the UNet or ResNet’s score function.  We chose the patch-size scale of the ELS machine to maximize this correlation at each separate time. Details can be found in our Appendix. We hope this constitutes a principled way to calibrate the time-dependent scale of the ELS machine.

---

### Decision · Program_Chairs · 2025-05-01

**Decision:**

Accept (oral)

**Comment:**

The paper received enthusiastic evaluations from all reviewers. Reviewers appreciate the design of experiments, theoretical derivations, and clear presentation. One of the reviewers suggested replacing the term creativity with another (more technical) term, e.g., generalization. After reading the paper, I share the reviewer’s perspective on this and encourage authors to consider this suggestion. Relevant references are appropriately included. This is a high quality work.